# Signed and unsigned reward prediction errors dynamically enhance learning and memory

Nina Rouhani[1]*, Yael Niv[2,3]

[1]Chen Neuroscience Institute, California Institute of Technology, Pasadena, United States; [2]Department of Psychology, Princeton University, Princeton, United States; [3]Princeton Neuroscience Institute, Princeton University, Princeton, United States

**Abstract** Memory helps guide behavior, but which experiences from the past are prioritized? Classic models of learning posit that events associated with unpredictable outcomes as well as, paradoxically, predictable outcomes, deploy more attention and learning for those events. Here, we test reinforcement learning and subsequent memory for those events, and treat signed and unsigned reward prediction errors (RPEs), experienced at the reward-predictive cue or reward outcome, as drivers of these two seemingly contradictory signals. By fitting reinforcement learning models to behavior, we find that both RPEs contribute to learning by modulating a dynamically changing learning rate. We further characterize the effects of these RPE signals on memory and show that both signed and unsigned RPEs enhance memory, in line with midbrain dopamine and locus-coeruleus modulation of hippocampal plasticity, thereby reconciling separate findings in the literature.

*For correspondence:
nrouhani@caltech.edu

## Introduction

The reward prediction error (RPE) is a canonical learning signal in reinforcement learning, updating stored information about the values of different experiences. This signal modulates dopaminergic firing from the midbrain, increasing dopamine release when rewards are better than expected, and decreasing its release when rewards are worse than expected ('signed RPE'; *Barto, 1995*; *Montague et al., 1996*). Over the course of learning, this dopaminergic RPE transfers from unpredictable reward outcome to the cue predicting the reward (*Schultz et al., 1997*). The resulting signed RPE at cue putatively supports an associative model ('Mackintosh model'; *Mackintosh, 1975*) where attention increases for cues that reliably predict reward. This signed RPE could also give rise to stronger memory traces, given that neural plasticity in the hippocampus – the key structure for episodic memory – is modulated by dopamine (*Lisman and Grace, 2005*; *Shohamy and Adcock, 2010*).

An alternative possibility is that RPE magnitude, regardless of its sign ('unsigned RPE'), enhances learning and memory for surprisingly good or bad outcomes. In fact, the Pearce-Hall model of learning (*Pearce and Hall, 1980*), which contradicts the Mackintosh model, posits that attention is enhanced for cues that are accompanied by surprise, that is, those that co-occur with large unsigned RPEs. The effects of unsigned RPEs are thought to be mediated by the locus coeruleus-norepinephrine system, which responds to unexpected changes in stimulus-reinforcement contingencies, regardless of the sign of the outcome (for a review, see *Sara, 2009*). Moreover, recent evidence points to the locus coeruleus (LC), which co-releases dopamine with norpeinephrine, as providing an alternative source of dopamine to the hippocampus, giving rise to hippocampal memories (*Kempadoo et al., 2016*; *Takeuchi et al., 2016*; *Wagatsuma et al., 2018*).

Albeit paradoxical, it is theoretically possible that both surprise (Pearce-Hall model) and predictability (Mackintosh model) modulate memory throughout learning (*Le Pelley, 2004*; *Beesley et al., 2015*), but in different ways, and through distinct neural mechanisms. Previously, we found that unsigned, but not signed, RPEs experienced at reward outcome boost learning and memory (*Rouhani et al., 2018*), consistent with work showing better memory for surprising events (*Greve et al., 2017*; *Antony et al., 2021*). There is also recent support for signed RPEs experienced at reward-predictive cue to enhance memory (*Jang et al., 2019*), reminiscent of work showing memory benefits during periods of high-reward anticipation (*Adcock et al., 2006*; *Murty and Adcock, 2014*; *Stanek et al., 2019*; *Wittmann et al., 2005*).

Accordingly, we hypothesized a signed-RPE effect on memory during the reward-predicting cue once participants had learned cue values, as well as an unsigned-RPE effect on memory during reward outcome throughout learning. We included two trial-unique images on every learning trial, one at cue and one at outcome, to dissociate the effects of the two RPEs on memory (*Figure 1*).

We characterized these effects in two experiments that each prioritized the influence of one of these RPE signals. In Experiment 1, participants experienced large unsigned RPEs brought on by periods of high outcome variance ('high' versus 'low variance' contexts) and reward-value change points (changes to the mean of the underlying reward distribution). We expected these large unsigned RPEs, experienced at reward outcome, to modulate learning rate and boost memory for events throughout learning (*Figure 1A–C*). In Experiment 2, in contrast, participants learned the values of two categories of cues, eliciting RPEs at cue as well as outcome (*Figure 1D–F*). Here, the underlying reward distribution associated with each category did not change, allowing for RPEs at cue (i.e. a relative value signal) to increase in magnitude with more experience with each category. We expected both signed RPEs at cue and unsigned RPEs at outcome to influence learning rate and memory for those events.

To assess differential effects of RPEs on instructed versus incidental memory, we ran two versions of Experiment 2. In the instructed version (and in Experiment 1), participants were explicitly prompted to memorize as they were told they would later choose between the items presented during learning and win their associated reward values again. We thus incentivized participants to associate the trial-unique items with their reward values during learning. In the incidental version, we removed this instruction, making memory for the trial-unique items completely unintentional.

We analyzed learning, memory, and choice data using complementary approaches. To understand the learning process, we compared computational models of learning that formalized different putative effects of signed cue RPEs and unsigned outcome RPEs on subjects' predictions of trial-by-trial cue values. To test the effects of signed and unsigned cue and outcome RPEs on memory performance, we used mixed-effects modeling (also used to analyze learning and later choice) and Bayesian hierarchical modeling.

## Results

### Learning results

#### Unsigned reward prediction errors at outcome and signed reward prediction errors at cue influenced learning rate

We first tested whether RPEs experienced during reward learning predicted empirical, trial-by-trial, learning rates. Learning rates were measured by comparing consecutive predictions for the same cue category, and dividing the difference in predictions by the empirical outcome prediction error experienced on the earlier of the two trials (see *Equation (1)* in 'Materials and methods'). We treated the unsigned RPE at reward outcome (*Figure 1B*, in blue) as a 'Pearce-Hall' signal, as it reflects how unpredictable the reward was. We found the unsigned RPE at reward outcome boosted learning rate in both experiments, thereby providing direct behavioral evidence for this 'Pearce-Hall' component on learning rate (mixed-effects linear regression, Experiment 1: $\beta$ = 0.10, $t$ = 6.39, p<0.001; Experiment 2: $\beta$ = 0.07 $t$=8.79, p<0.001; see *Figure 2—figure supplement 1* for empirical learning rates).

We treated the learned value difference between two reward-predictive cues in Experiment 2 as a 'Mackintosh' signal, as higher learned values for one cue versus the other implied better reward predictiveness. We refer to this value signal as a signed cue RPE (*Figure 1E*, red), as when there are

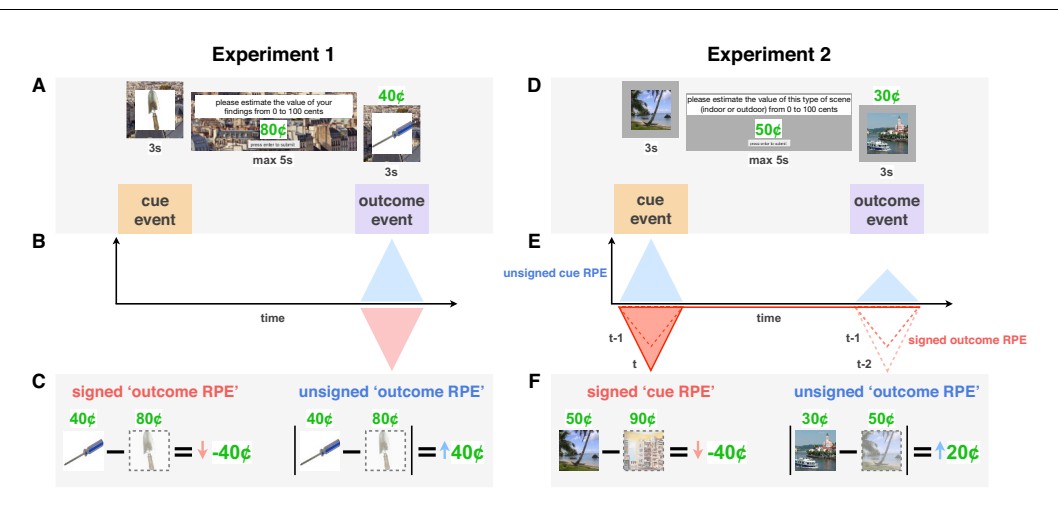

**Figure 1.** Reward prediction error (RPE) signals in a learning trial in Experiments 1 and 2. (**A,D**) Each trial was initiated by a reward-predicting cue represented by a trial-unique image. Participants were then asked to indicate how much that reward category was worth 'on average.' They then saw the reward outcome (a proportion of which they received) along with a second trial-unique image. In Experiment 1 (**A**), all images were of objects (single reward category), whereas in Experiment 2 (**D**), each trial included either two indoor or two outdoor scenes (two cue categories). (**B-F**) Theoretical RPE signals (**B,E**) and their calculation (**C,F**). Unsigned RPEs at outcome (in blue) were calculated by taking the absolute difference between the participant's value for that reward cue and the subsequent outcome. We expected this (putatively noradrenergic) unsigned signal to enhance memory for more surprising outcomes, which we tested in both Experiments 1 and 2. Signed RPEs at cue (**E**, in red) were calculated by taking the difference between the participant's predicted value for the current reward category (here, outdoor scenes) and their most recently predicted value of the other category (indoor scenes). We expected this (putatively dopaminergic) signed signal to boost memory for more valued events, that is, better memory the more positive the RPE. Prediction errors at outcome gradually transfer to cue through the learning process (**E**, dotted lines represent signed RPE in previous two trials, darker red indicates more recent trial).

several possible cues, the onset of a cue resolves the prediction for the current trial, and is accompanied by an RPE that reflects the signed difference between the current predicted reward and the average reward predicted before cue onset (*Niv and Schoenbaum, 2008*). We found that a signed cue RPE was anti-correlated with learning rate, potentially demonstrating stronger associative links and more stable values for more valuable cues ($\beta = -0.02$ $t = -2.31$, p=0.02). Critically, we found this effect even when controlling for any effect of the unsigned cue RPE on learning rate (which was not itself significant, $\beta = -0.01$ $t = -1.35$, p=0.18). This suggests that the cue RPE modulation of learning rate was not merely due to the greater learned separation between the two cue values, but specific to more stable updating for the high-valued cues.

## Learning behavior in the experimental conditions of Experiment 2

Experiment 2 involved four conditions in a between-participants 2 × 2 design. First, two learning conditions varied in difficulty due to different degrees of overlap between the reward distributions of the two categories. In the 40¢−60¢ condition, the means of the two reward categories were 40¢ and 60¢, with considerable overlap in the two reward distributions. In the 20¢−80¢ condition, on the other hand, the two means were 20¢ and 80¢, and there was no overlap between the two reward distributions. As expected, participants separated the values of two scene categories more in the 20¢−80¢ condition than in the 40¢−60¢ condition (*Figure 2B*), both in general and as a function of trial number during learning (mixed-effects linear regression, value separation as a function of learning condition: $\beta = 18.62$, $t = 25.87$, p<0.001; interaction between learning condition and trial number: $\beta = 5.40$, $t = 13.64$, p<0.001).

We also manipulated whether participants were intentionally or incidentally encoding the trial-unique scenes. We either instructed them to attend to the scenes in order to choose between them and win their associated reward later in the task ('instructed memory'), or we did not provide any

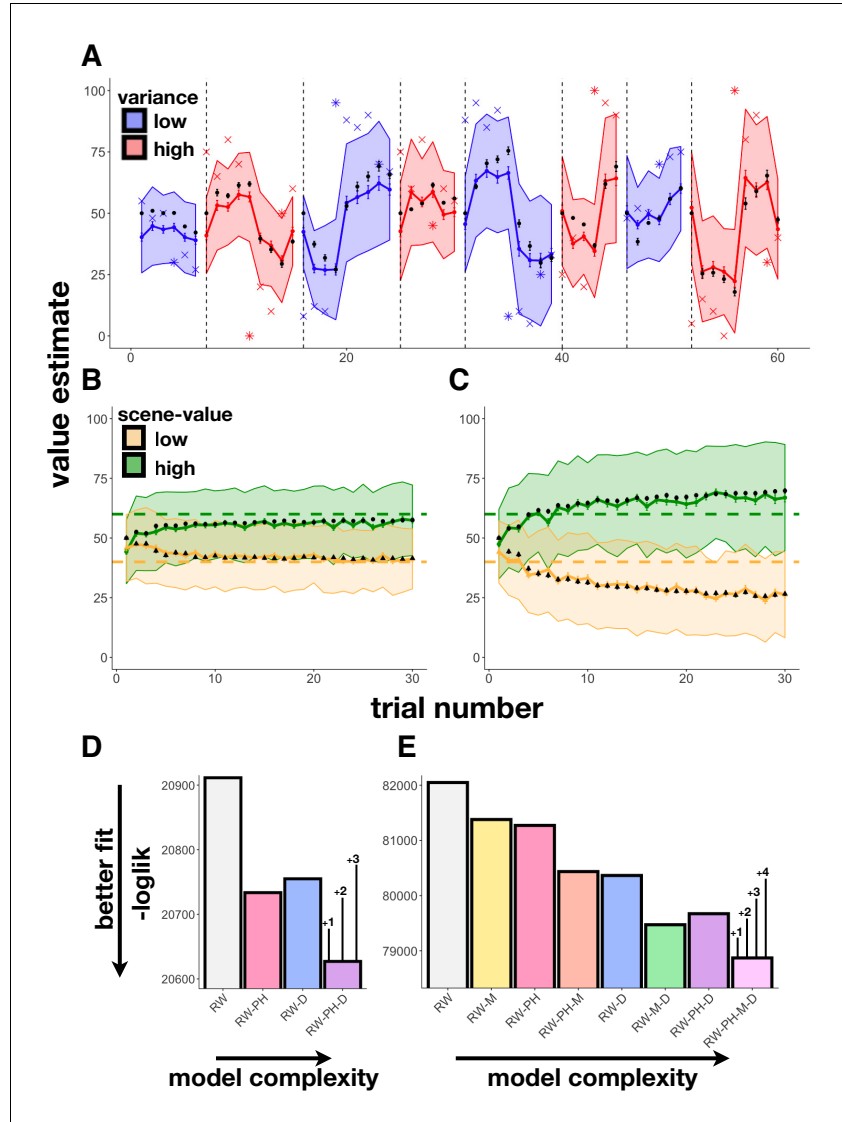

**Figure 2.** Learning behavior and modeling results. (**A**) Experiment 1 average participant value estimates as a function of trial number (blue and red lines represent two different outcome-variance contexts: blue = 'low variance' learning context, red = 'high variance' learning context; shading indicates 95% confidence intervals), and average predictions of the RW-PH-D model with SEM bars in black. Actual reward outcomes on each trial are indicated by x's, stars indicate a change-point trial. (**B-C**) Experiment 2 average participant value estimates for the two scene categories (green and yellow) as a function of trial number in the 40¢−60¢ condition (**B**; means of the two scene categories 40¢ and 60¢) and the 20¢−80¢ condition (**C**; average means 20¢ and 80¢), and average predictions of the RW-PH-M-D model in black. Actual rewards varied across subjects. Although each subject saw only 15 trials of each scene type (one of two scene-value categories on each trial), we pseudo-randomized the sequence of scene-value categories so that across participants, we had data for both categories on each trial. (**D-E**) Total negative log-likelihood scores across subjects for each of the models tested. Lower scores indicate better fit between model predictions and empirical data; bars on the winning model indicate the minimum difference needed for a significant difference between models in the likelihood-ratio test, given the number of extra parameters in the more complex model; 'RW': Rescorla-Wagner, 'PH': Pearce-Hall, 'M': Mackintosh, 'D': Decay. In Experiment 1 (**D**), the RW-PH-D model, which included a Pearce-Hall and a decay component, was the winning model. In Experiment 2 (**E**), the RW-PH-M-D, which additionally included a Mackintosh component, outperformed the other models.

The online version of this article includes the following figure supplement(s) for figure 2:

**Figure supplement 1.** Empirical learning rates in Experiments 1 and 2.

**Figure supplement 2.** Model validation simulations from Experiment 2.

instruction motivating remembering of the scenes ('incidental memory'). Although the instructions regarding value learning and prediction were identical for all participants, we did find overall differences in learning across the two memory-instruction conditions. Participants learned better and were more accurate in the instructed memory version: their estimates were closer to the actual underlying means of the scene categories and those estimates became more accurate over time (learning accuracy as a function of instructed-incidental memory: $\beta$ = 4.35, $t$ = 8.16, p<0.001; interaction between instructed-incidental memory and trial number: $\beta$ = 0.48, $t$ = 2.08, p=0.04).

Although we did not expect this difference in learning, it is possible that motivating the remembering of more valuable scenes led participants to attend more to learning those values as well. Additionally, we note that the instructed and incidental memory versions of the task were tested during different social climates, with the data for the incidental memory version collected during 2020's global pandemic, potentially accounting for the difference in learning performance. Interestingly, participants' estimates, and thus expectations for reward, were, on average, lower during the pandemic than before, demonstrating more pessimistic expectations overall ($\beta$ = −1.17, $t$ = −2.65, p=0.008).

## Reinforcement-learning models

To further determine how unsigned RPEs at reward outcome ('outcome RPEs') and signed RPEs at reward-predicting cue ('cue RPEs') influence learning, we modeled participants' trial-by-trial value estimates testing a series of reinforcement-learning models. We modeled Experiments 1 and 2 separately due to their different learning structures, and modeled all learning and memory conditions of Experiment 2 together since the learning instructions and structure were the same across all variants. We also performed and confirmed model recovery on simulated data to verify that our data can arbitrate between these models (see 'Model fitting and comparison' in 'Materials and methods').

### Experiment 1

We fit learning behavior in Experiment 1 using four models: a Rescorla-Wagner model with a fixed learning rate ('RW'), an RW-model with a Pearce-Hall (*Pearce and Hall, 1980*) component modulating learning rate ('RW-PH'), an RW-model with a decaying learning rate ('RW-D'), and a full model with both Pearce-Hall modulation and decay ('RW-PH-D'); see 'Materials and methods'. Note, we did not test models that included cue RPEs since there was a single reward category in this experiment. We found that the full model that included a Pearce-Hall component, which modulated learning rate by the unsigned outcome RPE, along with a decay, fit better than models without those components (likelihood-ratio tests, RW-PH-D vs. RW: $\chi^2$ (243) = 568.45, p<0.001; RW-PH-D vs. RW-PH: $\chi^2$ (162) = 212.87, p=0.005; RW-PH-D vs. RW-D: $\chi^2$ (81) = 255.71, p<0.001; *Figure 2A,D*, *Table 1*).

### Experiment 2

Here, participants experienced RPEs at both cue and outcome, allowing us to test the models above, as well as four models that included a Mackintosh-type component (denoted by 'M': 'RW-M', 'RW-PH-M', 'RW-M-D', 'RW-PH-M-D'; see 'Materials and methods'). The model that included all three tested modulators of learning rate—an unsigned RPE at outcome (Pearce-Hall component), a signed RPE at cue (Mackintosh component), and an exponential decay—predicted participant value estimates best (*Figure 2B,C,E*, *Table 1*). This model had a significantly better (i.e. lower) likelihood compared to other model as assessed by the likelihood-ratio test (RW-PH-M-D vs. RW: $\chi^2$ (2740)=6358.34, p<0.001; RW-PH-M-D vs. RW-M: $\chi^2$ (2055)=5017.65, p<0.001; RW-PH-M-D vs. RW-PH: $\chi^2$ (2055)=4803.86, p<0.001; RW-PH-M-D vs. RW-PH-M: $\chi^2$ (1370)=3130.53, p<0.001; RW-PH-M-D vs. RW-D: $\chi^2$ (1370)=2989.18, p<0.001; RW-PH-M-D vs. RW-M-D: $\chi^2$ (685)=1203.79, p<0.001; RW-PH-M-D vs. RW-PH-D: $\chi^2$ (685)=1603.55, p<0.001).

## Memory results by learning condition

To understand how cue and outcome RPEs affected memory for the trial-unique images, we analyzed memory results separately for Experiment 1 and the four conditions of Experiment 2 (instructed or incidental memory x learning difficulty). For each participant and each item tested, we calculated a memory score that combines memory accuracy (hit or miss) with confidence (from 1 =

**Table 1.** Model parameters and fit results.
'RW': Rescorla-Wagner, 'PH': Pearce-Hall, 'M': Mackintosh, 'D': Decay. Negative log-likelihood across participants for Experiment 1 (first row within each model) and Experiment 2 (second row within each model); 'd' refers to the difference in score between the tested model and the baseline fixed learning rate model ('RW'). Lower scores indicate better fit. In Experiments 1 and 2, models that included all tested components of learning rate performed best according to the likelihood-ratio test (which penalizes nested models for added parameters).

| Model | Parameters | -LL |
|---|---|---|
| RW | $\alpha$ | 20911.33 |
| | | 82049.33 |
| RW-PH | $\eta, \kappa$ | 20733.54 (d = -177.79) |
| | | 81272.09 (d = -777.24) |
| RW-M | $\eta, \gamma$ | 81378.98 (d = -670.35) |
| RW-D | $\eta, N, \lambda$ | 20754.96 (d = -156.37) |
| | | 80364.74 (d = -1684.58) |
| RW-PH-M | $\eta, \kappa, \gamma$ | 80435.42 (d = -1613.90) |
| RW-PH-D | $\eta, \kappa, N, \lambda$ | 20627.11 (d = -284.22) |
| | | 79671.93 (d = -2377.39) |
| RW-M-D | $\eta, \gamma, N, \lambda$ | 79472.05 (d = -2577.28) |
| RW-PH-M-D | $\eta, \kappa, \gamma, N, \lambda$ | 78870.16 (d = -3179.17) |

'guessing' to 4 = 'completely certain'), ranging from a 'completely certain' miss (1) to a 'completely certain' hit (8).

## High reward variance boosted memory for outcome events

Experiment 1 allowed us to test how reward variance modulates memory for cue and outcome events. In line with our previous work (*Rouhani et al., 2018*), we expected that the larger unsigned RPEs in a high-variance context would improve memory for related events, and therefore memory for high-variance items would be better overall. We found an interaction of cue versus outcome memory by variance condition, such that in the high-variance condition, there was a lower average memory score for cue events, and a higher average memory score for outcome events, compared to the low-variance condition ($\mu$ high-variance cue memory = 6.44, $\mu$ low-variance cue memory = 6.57, $\mu$ high-variance outcome memory = 5.79, $\mu$ low-variance outcome memory = 5.54; mixed-effects linear regression: $\beta = -0.37$, $t = -2.78$, p=0.005). Within the interaction, there was a significant difference in memory for outcome events ($\beta = -0.25$, $t = -2.09$, p=0.04) but not for cue events ($\beta = 0.12$, $t = 1.41$, p=0.16). This suggests a role for the high-variance context, characterized by larger unsigned RPEs, in boosting memory for outcome events.

## Memory for cue events increased with reward learning

We tested the effects of cue RPEs on memory by first comparing differences in cue memory in Experiment 1 and the instructed memory version of Experiment 2. In both experiments, participants were told they would have a chance to select among previously-seen items in a later phase, which encouraged explicit encoding of the items. Experiment 1 involved a single reward category and therefore did not elicit RPEs at cue. Frequent change points in the underlying reward distribution encouraged ongoing new learning. In Experiment 2, on the other hand, participants learned about two reward categories, evoking RPEs at cue, and the underlying reward distributions did not change, encouraging convergence of learning. We therefore predicted memory for cue items to be modulated by learning in Experiment 2 but not in Experiment 1. Since the additional monetary outcome accompanying the outcome image may, in general, interfere with encoding of the image, we also expected overall better memory for cue as compared to outcome images. We controlled for this nuisance effect in all our analyses.

In line with our predictions, in Experiment 1, we did not find cue memory to change relative to outcome memory throughout learning (mixed-effects linear regression, memory as a function of the interaction between event type (cue or outcome) and learning trial, $\beta$ = 0.009, $t$ = 0.13, p=0.90; *Figure 3A*), whereas in Experiment 2, memory for cue events improved throughout the experiment, relative to memory for outcome events, in both difficulty conditions (interaction between event type and learning trial, 40¢−60¢ condition: $\beta$ = −0.24, $t$ = −3.76, p<0.001; 20¢−80¢ condition: $\beta$ = −0.20, $t$ = −3.29, p<0.001; *Figure 3B–C*).

We next compared memory results between the instructed and incidental memory versions of Experiment 2. We did not find the increase in cue memory as a function of learning in the 40¢−60¢ learning condition (interaction between event type and learning trial: $\beta$ = 0.07, $t$ = 1.11, p=0.27; three-way interaction between event type, instructed-incidental memory and learning trial $\beta$ = 0.31, $t$ = 3.39, p<0.001; *Figure 3D*). However, we did replicate the increase in cue memory in the 20¢−80¢ condition (interaction between event type and learning trial: $\beta$ = −0.17, $t$ = −2.79, p=0.005; three-way interaction between event type, instructed-incidental memory and learning trial: $\beta$ = 0.04, $t$ = 0.41, p=0.68; *Figure 3E*).

To explain the difference in cue memory between the instructed and incidental memory versions of the 40¢−60¢ condition, we first note that learning to predict the outcome value was in general worse in the incidental memory version (see 'Learning behavior in the experimental conditions of Experiment 2', above). As the 40¢−60¢ condition was the more difficult learning environment, it is possible that worse learning prevented an average increase in cue memory as a function of learning. To test this hypothesis, we investigated whether individual differences in outcome-value learning predict differences in cue memory over learning. We computed learning performance for each participant by averaging the estimates of the last five trials of each scene category, and subtracting the average estimate of the low-value scene category from that of the high-value scene category. A larger, positive difference between the two scene categories indicates greater learned separation of the values of the two scene categories, whereas a smaller or negative difference indicates worse learning. To measure the individual increase in memory for events as a function of learning, we ran two mixed-effects models predicting memory as a function of trial number for (1) cue events and (2) outcome events, and extracted participant slopes from each model. We then tested whether individual learning performance predicted this change in cue or outcome memory over learning.

Overall, better individual learning predicted a greater increase in memory for cue events (linear regression: $\beta$ = 0.02, $t$ = 3.07, p=0.002), but not for outcome events ($\beta$ = 0.001, $t$ = 0.24, p=0.81; interaction between event type and learning performance: $\beta$ = −0.01, $t$ = −2.29, p=0.02). This relationship was stronger in the incidental memory task ($\beta$ = 0.02, $t$ = 3.61, p<0.001; see *Figure 3—figure supplement 1B*) than in the instructed one ($\beta$ = −0.004, $t$ = −0.61, p=0.55, *Figure 3—figure supplement 1A*; interaction between instructed-incidental memory and learning performance: $\beta$ = 0.03, $t$ = 2.61, p=0.009), and in the 40¢−60¢ condition, the more difficult learning task, relative to the 20¢−80¢ condition (40¢−60¢ condition: $\beta$ = 0.03, $t$ = 4.57, p<0.001; 20¢−80¢ condition: $\beta$ = 0.01, $t$ = 1.50, p=0.13; interaction between condition and learning performance: $\beta$ = −0.04, $t$ = −3.17, p=0.002). These results confirm that more learning led to a greater increase in memory for cue events. Furthermore, it suggests that the difference in results between the instructed and incidental memory versions of the 40¢−60¢ condition could be accounted for by worse learning performance. In this condition, there is a strong relationship between learning performance and learning-modulated cue memory: here, only a minority of participants who had learned to separate the values of the scene categories showed an increase in cue memory over learning, leading to an overall lack of effect at the group level.

## Memory results by reward prediction error

We investigated the effects of trial-by-trial reward prediction errors at cue and outcome in two ways, and modeled Experiment 1 and the instructed and incidental memory versions of Experiment 2 separately (see 'Materials and methods' for details). First, we used mixed-effects linear regression modeling to test the overall effects of RPEs on memory, including interactions between cue and outcome events, and then examined the effects of RPEs on cue and outcome memory separately. This resulted in three mixed-effects regression models for each experiment. We report model estimates and significance testing for these tests. Second, we ran three Bayesian hierarchical models (again, one model per experiment) including all RPEs as regressors and using a confound regressor to

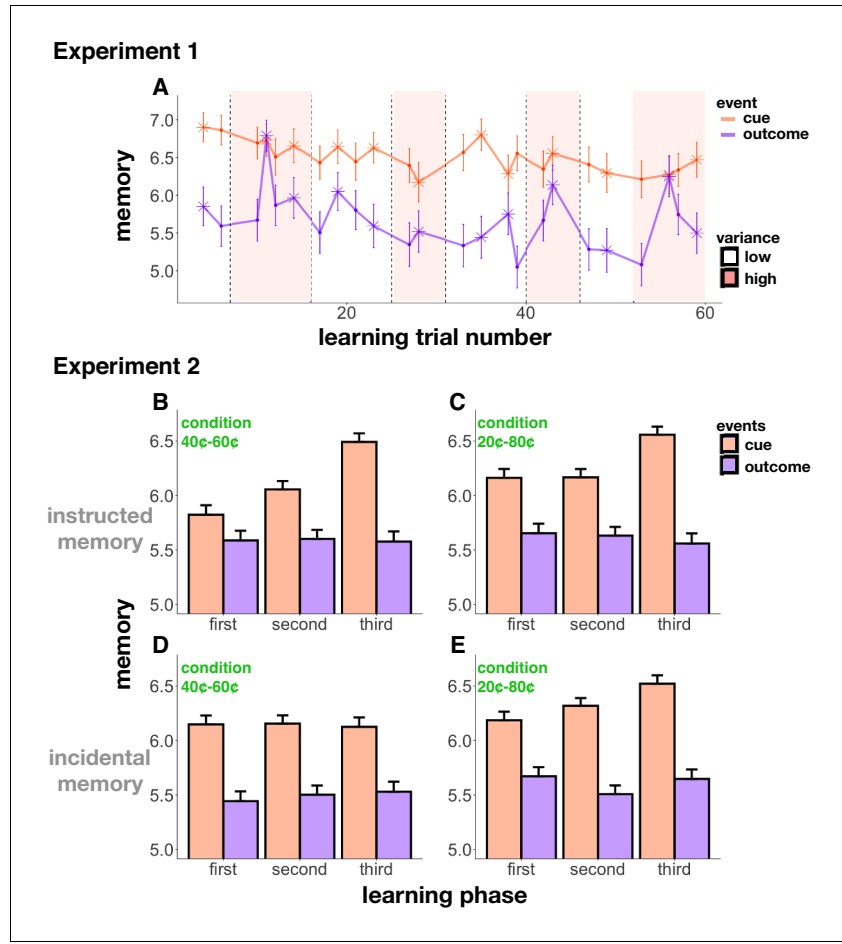

**Figure 3.** Memory accuracy across learning. (**A**) Experiment 1 memory score as a function of trial number; starred points indicate change-point events. Background shading indicates condition (low- or high-reward variance). Cue memory (in orange) was in general better than outcome memory (in purple; this effect was controlled for in all of our analyses). Cue memory did not change relative to outcome memory throughout learning. Reward change-points (starred) increased memory for the outcome event. (**B-E**) Experiment 2 memory scores in the 40¢−60¢ condition (**B,D**) and the 20¢−80¢ condition (**C,E**), as a function of 'learning phase' (first, second, and third bins of learning trials). In the instructed memory version of Experiment 2 (**B,C**), learning enhanced cue memory in both conditions, whereas in the incidental memory version, this enhancement only occurred in the 20¢−80¢ condition, the easier learning condition (**E**). Differences between the instructed and incidental memory versions of the 40¢−60¢ condition were related to differences in learning performance (see main text).

The online version of this article includes the following figure supplement(s) for figure 3:

**Figure supplement 1.** Individual differences in memory for cue and outcome events as a function of learning performance in the instructed (**A**) and incidental (**B**) memory versions of Experiment 2.

control for the, on average, better memory for cue events. We report the median (measure of centrality, 'M') and the high-density interval (measure of uncertainty, 'HDI') from the posterior parameter distributions generated by the Bayesian hierarchical model.

## Unsigned but not signed reward prediction errors at outcome enhanced memory

In all experiments, we found that unsigned outcome RPEs enhanced memory for outcome images (outcome memory as a function of |outcome RPE|, Experiment 1: M = 0.14, HDI [0.06, 0.23], $\beta$ = 0.19, $t$ = 3.51, p<0.001, *Figure 4A*; Experiment 2 - instructed memory: M = 0.14, HDI [0.05, 0.23], $\beta$ = 0.09, $t$ = 2.59, p=0.009, *Figure 4D*; Experiment 2 - incidental memory: M = 0.13, HDI [0.06, 0.19], $\beta$ = 0.11, $t$ = 3.01, p=0.003, *Figure 4D*).

Unsigned outcome RPEs in Experiment 1 resulted from either reward variance or changes to the underlying distribution of the mean. We found separate effects of RPEs due only to reward variance and those due to change points on increasing memory for outcome images (outcome memory as a function of variance RPEs: $\beta$ = 0.14, $t$ = 2.38, p=0.02; outcome memory as a function of change-point RPEs: $\beta$ = 0.24, $t$ = 2.08, p=0.04).

In Experiment 1, we also found unsigned outcome RPEs to boost memory for cue images from that same trial, but with a median parameter estimate half the size of that for outcome images (cue memory as a function of |outcome RPE|: M = 0.08, HDI [−0.01, 0.16], $\beta$ = 0.10, $t$ = 2.29, p=0.02, *Figure 4A*). We did not find this effect in any of the conditions of Experiment 2, where unsigned outcome RPE enhanced memory only for outcome images (Experiment 2 - instructed memory, cue memory as a function of |outcome RPE|: M = −0.02, HDI [−0.11, 0.06], $\beta$ = −0.01, $t$ = −0.46, p=0.64, interaction between event type (cue or outcome event) and |outcome RPE|: $\beta$ = 0.09, $t$ = 2.06, p=0.04; Experiment 2 - incidental memory, cue memory as a function of |outcome RPE|: M = −0.004, HDI [−0.07, 0.07], $\beta$ = 0.006, $t$ = 0.20, p=0.84, interaction between event type and |outcome RPE|: $\beta$ = 0.09, $t$ = 1.98, p=0.05, *Figure 4D*).

As expected based on previous work (*Rouhani et al., 2018*), we did not find any influence of signed outcome RPEs on memory for cue or outcome images in either experiment (*Figure 4B,F*; cue memory as a function of signed outcome RPE, Experiment 1: M = −0.02, HDI [−0.11, 0.06], $\beta$ = −0.05, $t$ = −1.04, p=0.30; Experiment 2 - instructed memory, M = 0.002, HDI [−0.07, 0.07], $\beta$ = −0.02, $t$ = −0.66, p=0.51; Experiment 2 - incidental memory, M = −0.01, HDI [−0.08, 0.05], $\beta$ = −0.003, $t$ = −0.08, p=0.93; outcome memory as a function of signed outcome RPE, Experiment 1: M = −0.0007, HDI [−0.09, 0.08], $\beta$ = −0.004, $t$ = −0.07, p=0.95; Experiment 2 - instructed memory, M = −0.03, HDI [−0.10, 0.03], $\beta$ = −0.05, $t$ = −1.46, p=0.15; Experiment 2 - incidental memory, M = −0.05, HDI [−0.11, 0.02], $\beta$ = −0.06, $t$ = −1.61, p=0.11).

## Reward prediction errors at cue enhanced memory

Prediction errors at cue were elicited only in Experiment 2. In the instructed memory version of Experiment 2, we found that signed cue RPEs (i.e. the signed difference between the participant-reported value of the current cue and the most-recently reported value of the alternative cue) boosted memory for both cue and outcome events, such that memory of higher (relative) value scenes was better than that for lower (relative) value scenes (cue memory as a function of signed cue RPE: M = 0.08, HDI [0.01, 0.15], $\beta$ = 0.08, $t$ = 2.64, p=0.008; outcome memory as a function of signed cue RPE: M = 0.06, HDI [−0.01, 0.12], $\beta$ = 0.07, $t$ = 2.23, p=0.03; *Figure 4E*). In the incidental version, we replicated the effect of cue RPE on memory for cue images, but not on memory for outcome images (cue memory as a function of signed cue RPE: M = 0.10, HDI [0.04, 0.17], $\beta$ = 0.09, $t$ = 3.04, p=0.002; outcome memory as a function of signed cue RPE: M = 0.03, HDI [−0.04, 0.09], $\beta$ = 0.03, $t$ = 0.80, p=0.42; *Figure 4E*).

We found a separate effect of unsigned cue RPE on memory, such that the more participants had separated the values of the two scene categories (i.e. the more they had learned), the better their memory for scene images. This effect was evident in overall memory in the instructed memory version of Experiment 2 (memory as a function of |cue RPE|: $\beta$ = 0.07, $t$ = 2.49, p=0.01), however, when quantifying the effect for cue and outcome memory separately, each of them was only trending to significance (cue memory as a function of |cue RPE|: M = 0.09, HDI [0.01, 0.16], $\beta$ = 0.07, $t$ = 1.88, p=0.06; outcome memory as a function of |cue RPE|: M = 0.06, HDI [−0.01, 0.13], $\beta$ = 0.08, $t$ = 1.82, p=0.07; *Figure 4C*). In the incidental version of Experiment 2, unsigned cue RPEs significantly increased memory for cue images (cue memory as a function of |cue RPE|: M = 0.12, HDI [0.05, 0.18], $\beta$ = 0.10, $t$ = 2.93, p=0.003), but not outcome images (outcome memory as a function of |cue RPE|: M = 0.02, HDI [−0.05, 0.08], $\beta$ = 0.01, $t$ = 0.30, p=0.76; interaction between event type and |cue RPE|: $\beta$ = −0.10, $t$ = −2.22, p=0.03; *Figure 4C*). The unsigned cue RPE's increase of, in particular, memory for cue images suggests an additional mechanism for the learning effects described above.

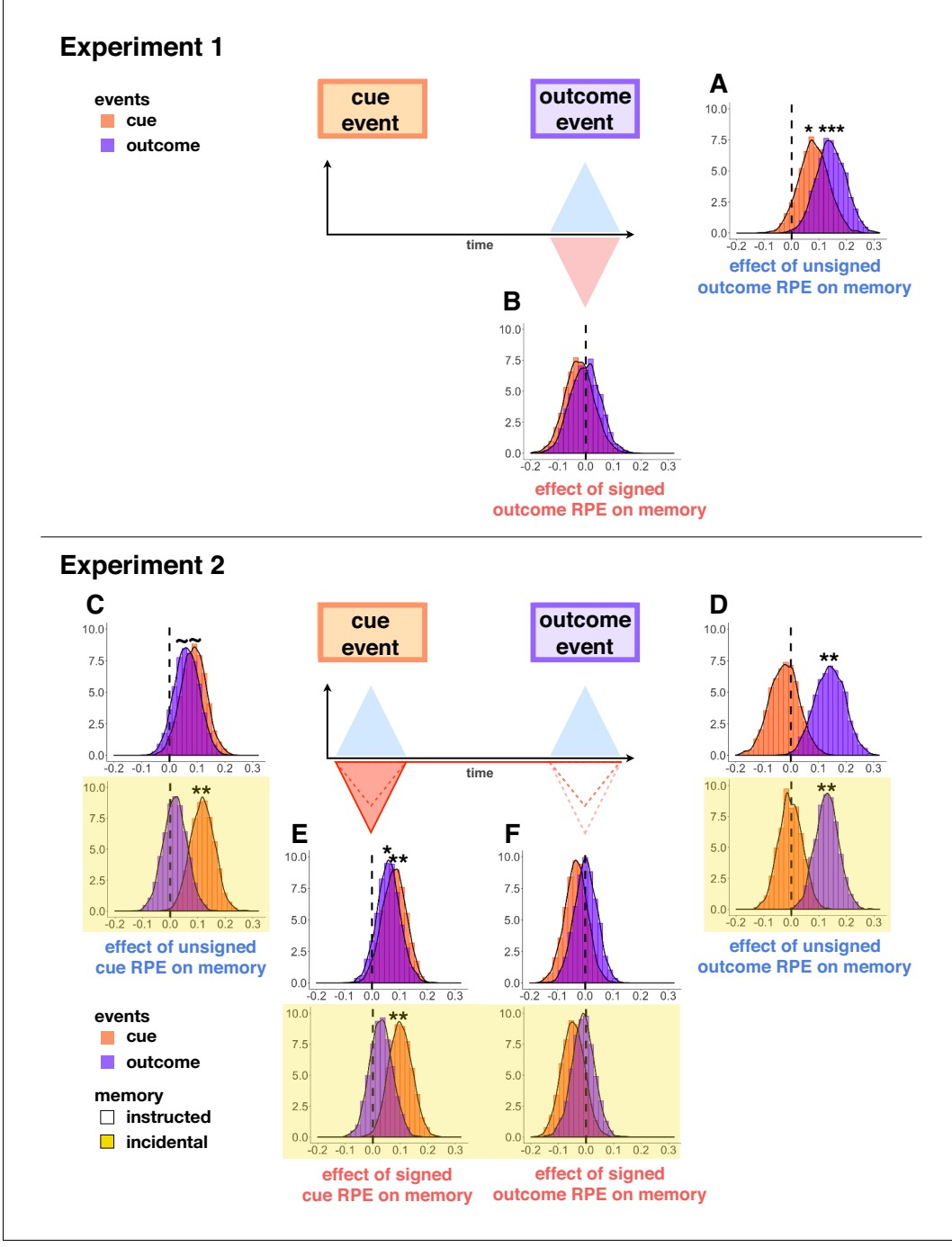

**Figure 4.** Parameter distributions from hierarchical Bayesian models of memory in Experiment 1 (A–B) and Experiment 2 (C-F, yellow background indicates incidental memory version). Distributions significantly above or below zero indicate an effect, black stars indicate significance: p<0.1~, p<0.05*, p<0.01**. Unsigned outcome RPEs (A,D) increased memory for outcome events, whereas signed outcome RPEs (B,F) did not. Signed cue RPEs (E) boosted memory for the cue item, and also enhanced memory for the outcome item in the instructed memory version. Unsigned cue RPEs (C) additionally, and separately, enhanced memory for cue events in the incidental memory version of the task; this effect was trending in increasing cue and outcome events in the instructed memory task.

## Outcomes and values did not predict memory without reward prediction error

To rule out alternative modulators of memory, we also tested the effect of the trial-by-trial reward outcomes and value estimates. We did not find reward outcome, in of itself or in an interaction with cue or outcome event, to predict memory in any of the experiments (memory as a function of reward outcome, Experiment 1: $\beta = -0.03$, $t = -0.68$, p=0.50, interaction between event type and reward outcome: $\beta = 0.07$, $t = 1.07$, p=0.29; Experiment 2 - instructed memory: $\beta = 0.03$, $t = 0.91$, p=0.36, interaction between event type and reward outcome: $\beta = 0.02$, $t = 0.55$, p=0.58; Experiment 2 - incidental memory: $\beta = 0.05$, $t = 1.51$, p=0.13, interaction between event type and reward outcome: $\beta = -0.04$, $t = -0.98$, p=0.33).

When testing the effect of participant value estimates on memory, we similarly did not find participant value estimates, in of themselves or in an interaction with cue or outcome event, to predict memory in Experiment 1 (memory as a function of value estimate, $\beta = -0.007$, $t = -0.15$, p=0.88, interaction between event type and value estimate: $\beta = -0.004$, $t = -0.06$, p=0.95). In Experiment 2, the value estimates were strongly correlated with the signed RPE at cue ($r > 0.80$), therefore we could not enter them into a single model. When tested alone, value estimates were trending in enhancing memory in the instructed memory version of Experiment 2 (memory as a function of value estimate, $\beta = 0.06$, $t = 1.89$, p=0.06, interaction between event type and value estimate: $\beta = 0.05$, $t = 1.19$, p=0.24), and enhanced memory in the incidental memory version (memory as a function of value estimate, $\beta = 0.08$, $t = 2.58$, p=0.01, interaction between event type and value estimate: $\beta = -0.03$, $t = -0.62$, p=0.53). However, given that value estimates did not influence memory when there was no prediction error at cue (Experiment 1), we believe the parsimonious interpretation is that this mnemonic modulation is attributable to prediction errors rather than value estimates alone.

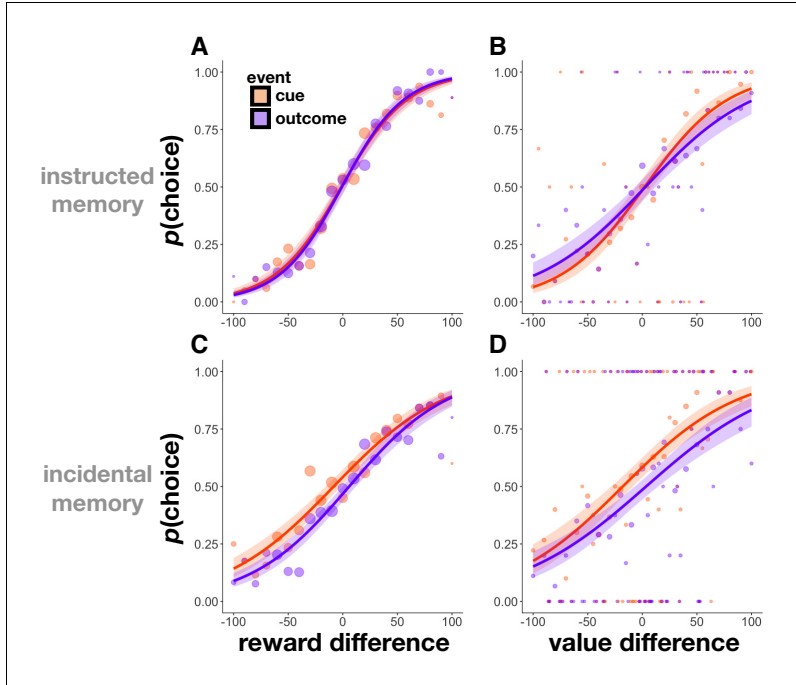

**Figure 5.** Choice probability as a function of rewards and values in Experiment 2. (A,C) Choice probability as a function of the difference in reward outcomes between two cue or two outcome items in Experiment 2. Participants were more likely to choose cue and outcome items that had been associated with higher reward outcomes in both the instructed (A) and incidental memory (C) versions of the task. (B,D) Choice probability as a function of the difference in value between two cue or two outcome items. Participants were more likely to choose cue and outcome items that they had associated with a more valuable scene category (relative to the other scene category) at the time of encoding in both the instructed (B) and incidental memory (D) versions of the task. Size of circles reflects the size of that sample. Choice was fit using a logistic function, and shaded regions reflect 95% confidence intervals.

## Choice results

At the end of both experiments, we asked participants to make choices between previously seen items. Choices were between items that were either (1) two cue or two outcome items from different trials (Experiment 2 only), or (2) a cue and an outcome item from a single trial, associated with a single value estimate and reward outcome (both experiments). No outcomes were presented after choices were made.

Recall that in Experiment 1 and the instructed memory version of Experiment 2, participants were told in advance to pay attention to the images and their outcomes as they would later have a chance to make choices between them and win their associated reward again. These instructions encouraged participants to encode the images along with their reward value. In the incidental version of Experiment 2, however, no such preview was provided, and participants were not given any incentive to encode the images nor their associated values or rewards.

### Outcomes and values increased choice

Choices between pairs of images associated with different reward outcomes and values were tested in Experiment 2. As expected, in the instructed memory version, participants preferred both cues and outcomes associated with higher rewards (mixed-effects logistic regression, choice as a function of reward difference: $\beta = 1.59$, $z = 20.13$, p<0.001; *Figure 5A*), and preferred images for which they had reported higher subjective value when controlling for the effect of reward outcome on choice (choice as a function of value difference: $\beta = 0.92$, $z = 7.58$, p<0.001; *Figure 5B*). Interestingly, both effects were replicated in the incidental memory version of the task, despite not motivating participants to encode nor associate images with their values and reward outcomes (choice as a function of reward difference: $\beta = 1.21$, $z = 12.28$, p<0.001; *Figure 5C*; choice as a function of value difference, controlling for reward difference: $\beta = 1.03$, $z = 7.84$, p<0.001; *Figure 5D*).

### Signed RPEs at outcome biased choice

The above results confirmed that participants associated both the cue and outcome event with their value of that category, as well as with the specific reward outcome on that trial, even when they were not instructed to do so. We also asked participants to choose between cue and outcome items from the same trial – two items that had the same associated value and reward outcome. In all experiments, we found that participants were more likely to prefer the outcome event the more positive the outcome RPE, and to prefer the cue event the more negative the outcome RPE (mixed-effects logistic regression, choice for outcome event as a function of signed outcome RPE,

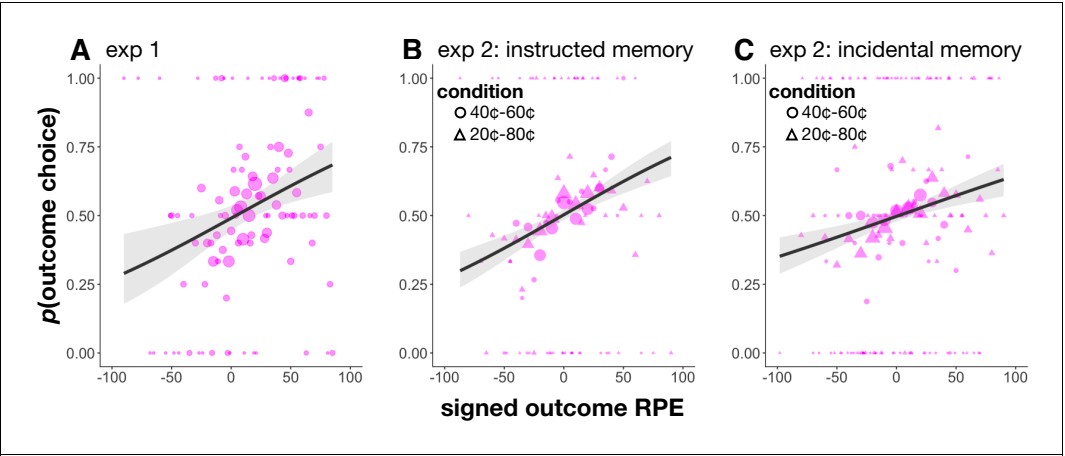

**Figure 6.** Choice between cue and outcome items from a single trial in Experiment 1 (**A**), the instructed memory version of Experiment 2 (**B**), and the incidental memory version of Experiment 2 (**C**). As these items were associated with the same reward outcome and value, we would not expect preference for either item. In all cases, participants preferred the outcome item on trials with a more positive outcome RPE and the cue item on trials with a more negative outcome RPE. Size of circles reflects the size of that sample. Choice was fit using a logistic function, and shaded regions reflect 95% confidence intervals.

Experiment 1: $\beta$ = 0.27, $z$ = 3.27, p=0.001, *Figure 6A*; Experiment 2 - instructed memory: $\beta$ = 0.23, $z$ = 5.70, p<0.001, *Figure 6B*; Experiment 2 - incidental memory: $\beta$ = 0.16, $z$ = 4.30, p<0.001, *Figure 6C*). Moreover, each one of these effects held when controlling for the magnitude of the outcome RPE, that is the unsigned outcome RPE (which boosted memory for outcome images, see 'Unsigned but not signed reward prediction errors at outcome enhanced memory', above), thus suggesting this preference may not be driven by better memory for large RPE events. Therefore, although signed outcome RPEs did not modulate memory, they did predict subsequent choice, pointing to a hedonic component of the signed RPE in shaping preference.

## Discussion

We found that distinct reward prediction error (RPE) signals, one occurring at cue and one at outcome, dynamically influenced learning and memory for those events. Drawing on classic associative models of attention (*Pearce and Mackintosh, 2010*), we found that an unsigned RPE at reward outcome modulated trial-specific learning rate, consistent with a Pearce-Hall model of learning that assumes more attention – and therefore more learning – for cues associated with unpredictable outcomes (*Pearce and Hall, 1980*). Similarly, a signed RPE at reward cue also modulated learning rate, consistent with a *Mackintosh, 1975* model of learning that assumes enhanced attention to reward-predicting cues. Reinforcement learning models that included both modulatory components predicted behavior better than models without those attentional components.

RPE signals at cue and outcome also enhanced memory for associated events. In Experiment 1, participants learned the value of a single reward category while experiencing large unsigned RPEs at outcome due to high (versus low) levels of outcome variance and unexpected changes in the mean of the underlying reward distribution ('change points'). We found that unsigned RPEs at reward outcome improved memory for trial-unique scenes accompanying both the cue and outcome, most prominently the latter.

In Experiment 2, participants learned the value of two reward categories (designated by indoor and outdoor trial-unique scenes), which meant that they experienced RPEs both at the time of the cue (as they could not predict which category would be offered on any given trial) and at the time of the reward outcome. Unlike Experiment 1, where memory for the cue event remained relatively stable throughout the task, in Experiment 2, memory for the cue event (but not for the outcome event) increased with learning. This mnemonic increase for cue events was supported by the gradual buildup of a signed RPE at cue, which enhanced memory for more valued reward cues, as well as an unsigned RPE at cue that benefited memory the more participants had separated the values of the two reward categories (i.e. the more they had learned). Furthermore, we found that individual learning performance predicted the increase in cue memory over learning, providing another link between stronger reward expectations and enhanced memory for cue events.

Similarly to Experiment 1, we again found that unsigned RPEs at reward outcome boosted memory for outcome events in Experiment 2, although we did not find them to influence memory for cue events. Importantly, our results were similar in both versions of Experiment 2, one that instructed memory for the images and their values and one where memory was completely incidental, confirming that the mnemonic benefits of RPEs are not a result of strategic encoding. Thus, our findings can be directly related to previous work characterizing implicit, incidental encoding during reward learning, as well as studies of animal conditioning in which encoding is not explicitly instructed (*Duszkiewicz et al., 2019*).

Last, in a choice test administered at the end of the experiment, participants preferred both cue and outcome scenes that had been associated with higher reward outcomes and more valued scene categories. This result was obtained when participants had been explicitly instructed to associate the events with their reward outcomes and values (instructed memory task) and when these associations were not instructed (incidental memory task). Additionally, when choosing between the cue and outcome scenes of a single trial (i.e. two scenes associated with the same reward and value), higher signed RPEs at outcome, which we did not find to modulate memory, nevertheless led to 'irrational' preference for the outcome event.

## Reward prediction errors dynamically modulated learning rate

We compared different reinforcement-learning models that included the contribution of attentional components in modulating learning rate on a trial-by-trial basis. We focused on attention's effect in enhancing overall learning rate, departing from classic paradigms that investigate the allocation of attention (or learning resources) between competing stimuli presented simultaneously. That is, we presented one stimulus at a time. Nevertheless, our experimental design allowed us to model and test the amount of learning for each stimulus on each trial, and investigate its relationship to RPEs. Empirically, we found that large unsigned RPEs boosted learning rate, in line with a Pearce-Hall model of attention for learning (*Pearce and Hall, 1980*), and previous work (*Rouhani et al., 2018*; *Nassar et al., 2010*). Model comparison suggested that modulating learning rate according to unsigned outcome RPEs fit learning behavior better than models without this modulation.

We also tested the influence of a Mackintosh-like attention component, which contrary to the Pearce-Hall model, predicts a change in attention for more valuable and predictive cues. We modeled the Mackintosh signal as the (signed) difference in learned value between the reward-predicting cues. We found that this signed cue RPE decreased empirical learning rates, meaning that values estimated for higher valued cues were more stable in light of unpredicted outcomes than for lower valued cues. This effect was not due to an overall decrease in learning rate after having learned the values of the two categories, as we did not find a significant effect of unsigned cue RPE on learning rate; instead, participants demonstrated more stable learning for higher-valued cues. A computational model that updated learning rate according to this signed RPE at cue, in addition to the unsigned RPE at outcome, provided the best fit to behavior, of all the models we tested.

In order to observe the effects of a building-up of RPE at cue onset, Experiment 2, unlike Experiment 1, did not include shifts in the underlying reward distribution of the categories. Future studies could examine how this Mackintosh signal changes with shifting predictions, where a change in the underlying mean of the rewards increases learning rate (*Vaghi et al., 2017*).

## Reward prediction error at cue benefited memory for cue events

Experiment 2, which included RPEs at cue, allowed us to test whether a putative (signed) dopaminergic RPE, which moves from reward outcome to the cue predicting reward over learning (*Barto, 1995*; *Montague et al., 1996*; *Schultz et al., 1997*), enhances memory for cue events. As predicted, we found an incremental increase in memory for cue events (and not outcome events) throughout learning. Moreover, we found this increase to be supported by a signed RPE at cue that boosted memory for cue events. That is, as learning progressed, cues that were more valuable (and therefore elicited a larger signed RPE at cue) were better remembered. This finding is consistent with previous work showing better memory for cues associated with future higher rewards (*Stanek et al., 2019*; *Jang et al., 2019*). As in our prior work (*Rouhani et al., 2018*), we did not find signed RPEs at reward outcome to modulate memory for any event, but we note that such an effect has been demonstrated in paradigms outside of reinforcement learning (*Marvin and Shohamy, 2016*; *Ergo et al., 2019*; *De Loof et al., 2018*; *Ergo et al., 2020*) and for adolescents, but not adults, in reinforcement learning (*Davidow et al., 2016*).

Along with the effect of signed cue RPE, the increase in cue memory over learning was additionally and separately supported by an unsigned RPE at cue, which improved memory for both high-value cues and low-value cues as participants learned to separate the values of the two reward categories. Although we consider the signed RPE at cue to reflect a Mackintosh-type attention signal (*Mackintosh, 1975*), in our paradigm, larger unsigned RPEs at cue also demonstrated that greater reward predictiveness strengthens encoding of those cue events.

Our findings further shed light on whether this mnemonic enhancement for cue events was due to cue values or cue RPE. We could not distinguish the effects of participant value estimates versus cue RPEs in Experiment 2, as the measures were highly correlated. However, in Experiment 1, where there were no RPEs experienced at cue, we did not find value estimates to modulate memory. We therefore experimentally dissociated the effects of cue values and RPEs on memory by showing that an increase in cue memory over learning requires the build-up of an RPE signal, which in Experiment 2 was elicited by learning about more than one reward category.

We also investigated whether individual differences in learning performance were related to this increase in cue memory over learning. More specifically, we tested whether value separation of the

two reward categories at the end of learning predicted individual changes in cue memory as a function of trial number during learning (the estimated slope of the effect). We indeed found learning performance to predict greater increases in cue memory, but not outcome memory, over learning. This relationship was strongest in the experimental condition associated with the worst learning performance (40¢−60¢, incidental memory condition). This was the only condition in which we did not find an overall increase in cue memory throughout learning, suggesting that this null result may be in part driven by worse learning in this condition, while better learners were still showing the effect. This analysis complemented our within-subjects approach by providing a between-subjects link between learning performance and increasing memory for cue events.

## Unsigned reward prediction error at outcome boosted memory throughout learning

We replicated previous results showing better memory for images associated with high unsigned RPEs at reward outcome, due to either outcome variance (*Rouhani et al., 2018*) or reward change-points (*Rouhani et al., 2020*). We found this mnemonic benefit of outcome RPEs to be particularly strong for events experienced at the time of the (surprising) outcome. Although unsigned outcome RPEs enhanced memory for cue events as well (see Experiment 1), this effect was weaker than that for outcome events, and did not replicate in Experiment 2. Critically, unsigned outcome RPEs benefited memory in both instructed and incidental memory conditions. We therefore hypothesize that, regardless of any explicit encoding strategy, increased attention due to large unsigned RPEs during reward outcome engages the locus coeruleus (LC), which co-releases norepinephrine and dopamine to modulate hippocampal plasticity (*Kempadoo et al., 2016*; *Takeuchi et al., 2016*; *Wagatsuma et al., 2018*). Although we only tested the effects of reward prediction errors, we speculate that any large prediction-error event would engage this putative neural mechanism to modulate memory, as supported by work showing mnemonic enhancement for large prediction-error events during fear conditioning (*Kalbe and Schwabe, 2019*).

Of note, we did not find effects of reward outcome on memory in any of our experiments, showing that the reward outcome by itself did not lead to the strategic prioritization of more rewarding events in the instructed memory version of our experiments, nor did it lead to the incidental encoding of those events. These results are consistent with reported null effects of reward outcome on immediate memory (*Murty et al., 2016a*). Indeed, reward effects on incidental memory typically emerge when tested after consolidation, a process in which rewarding events, and their associates, are prioritized in memory (*Braun et al., 2018*; *Stanek et al., 2019*; *Patil et al., 2017*). In our paradigm, encoding and recognition memory were separated by only a short delay, leaving open the possibility that reward effects on cue and outcome events would emerge after consolidation.

## Instructed versus incidental memory effects

In both Experiment 1 and the instructed memory version of Experiment 2, we incentivized participants to associate trial-unique images with their reward outcome: in the initial instructions, participants were told that they would later have an opportunity to choose between the images and win their reward outcomes again. These instructions may have encouraged participants to strategically encode the images, similar to work investigating explicit memory strategies for remembering rewarding events (e.g. *Hennessee et al., 2019*). Such explicit strategies may use different neural mechanisms than those motivating our questions, namely, the dopaminergic and noradrenergic modulation of hippocampal memories characterized in animal conditioning paradigms.

We addressed this potentially critical difference by running a version of Experiment 2 where memory for the images, as well as their association with the reward values and outcomes, were completely incidental. Here, we replicated our main results: signed and unsigned RPEs at cue enhanced memory for cue events, and unsigned RPEs at outcome enhanced memory for outcome events. Unlike the instructed memory version of Experiment 2, we did not find the signed RPE at cue to additionally increase memory for outcome events. This difference could potentially reflect a more explicit encoding of the outcome event for the later choice task in the instructed memory version, as participants were told that cue and outcome events from a single trial were associated with the same value and outcome. However, it is worth noting that a good explicit strategy would have

been to remember the most rewarding events, yet, reward outcome did not modulate memory in any of our experiments. We therefore conclude that our RPE effects do not rely on explicit memory strategies and are likely to be unintentional, connecting our findings to literature investigating incidental memory during reward learning.

We also found that learning was, on average, worse in the incidental memory compared to the instructed memory version of Experiment 2. We did not anticipate this difference since the only instructions we changed were related to the final choice task. However, it is possible that motivating participants to associate the images with their reward values and outcomes additionally encouraged them to learn and better separate the average values of the two reward categories. Alternatively, participant pool and motivation may be related to this difference, as we ran the incidental memory task during 2020's global pandemic. As a potential signature for decreased motivation in the 2020 participant pool, we found participants to provide lower value estimates, indicating greater pessimism when anticipating rewards. Nevertheless, this greater variability in learning performance in the incidental memory task did allow us to demonstrate that individual differences in learning predicted the degree to which cue memory increased over learning (see 'Reward prediction error at cue benefits memory for cue events', above).

## Interactions between reinforcement learning and memory systems

Although we did not measure neural activity in this study, distinguishing the mnemonic effects of signed and unsigned RPEs in the brain may be fruitful in characterizing two distinct memory mechanisms. As mentioned, one dominant hypothesis is that dopaminergic midbrain signals convey signed RPEs to target areas (*Barto, 1995*; *Montague et al., 1996*; *Schultz et al., 1997*); Less well accepted, but also quite dominant is the idea that unsigned RPEs increase noradrenergic (as well as dopaminergic) firing from the LC (*Takeuchi et al., 2016*; *Kempadoo et al., 2016*; *Wagatsuma et al., 2018*). Recent work makes predictions about how these distinct mechanisms may differentially influence memory (*Hauser et al., 2019*). Midbrain dopamine initiates 'behavioral activation' (*Clewett and Murty, 2019*), such as increased vigor during periods of reward anticipation (*Niv et al., 2007*), which is thought to promote the integration of higher-order representations, like value formation, giving rise to semantic memories (*Duszkiewicz et al., 2019*). The LC-norepinephrine system, on the other hand, is thought to promote selectivity for salient events such as (positively or negatively) surprising outcomes, giving rise to distinctive, episodic memories (*Duszkiewicz et al., 2019*). Although our findings suggest that RPEs act on both episodic (high-confidence recognition) and semantic (value formation) memory, distinguishing the effects these RPE signals may have on other features of episodic and semantic memory is an important avenue for future research (*Greve et al., 2019*).

In this paradigm, we did not dissociate the effects of cue RPE versus reward anticipation on memory (for an experiment that does this, see *Stanek et al., 2019*). However, based on our findings and the above mapping to neural substrates, we predict that phasic signed RPEs at cue would initiate and enhance a sustained (potentially ramping) period of reward anticipation, leading to memory benefits for ensuing events, regardless of their exact timepoint. Iigaya and colleagues recently offered such a computational model whereby RPEs amplify anticipatory value (i.e. the 'pleasure of savoring'; *Iigaya et al., 2019*). They further suggested a neural circuit whereby the hippocampus – tracking unsigned RPEs at outcome – enhances the functional coupling between the dopaminergic midbrain (encoding the signed RPE at outcome) and the ventromedial prefrontal cortex (encoding anticipatory value) to boost reward anticipation. The authors speculate that the cognitive imagining of future rewards may drive hippocampal orchestration of reward anticipation. It is, however, unclear whether hippocampal activation here reflects greater engagement in retrieval processes (supporting the mental simulation of future rewards) or encoding processes, consistent with previous work showing better memory for events experienced during reward anticipation (*Stanek et al., 2019*; *Murty and Adcock, 2014*; *Wittmann et al., 2005*). Future work should identify the dynamics of hippocampal encoding and retrieval states (*Hasselmo et al., 2002*; *Duncan et al., 2012*; *Bein et al., 2020*) over the period of reward anticipation.

In our experiments, we found a collaborative interaction between reinforcement learning and episodic memory systems: more rewarding cues and more surprising outcomes were prioritized in memory, thereby promoting adaptive behavior. Nonetheless, in other paradigms, these two systems have been shown to compete for processing resources: compromised feedback-based learning has

been associated with enhanced episodic memory, both behaviorally and neurally (*Foerde et al., 2013*; *Wimmer et al., 2014*). In fact, *Wimmer et al., 2014* showed that better memory for reward-predicting cues was associated with weaker striatal RPEs at reward outcome. In our experiments, we did not find such effects of signed RPEs at reward outcome on memory. However, there are several notable differences between our task and that of Wimmer and colleagues: we tested Pavlovian (passive) learning, not instrumental learning (choice); we presented one cue at a time, rather than two competing cues on every trial; and we tested for memory immediately after learning, rather than 24 hr later (that is, our test did not reflect consolidation effects).

In our Pavlovian paradigm, participants' actions (i.e. their estimates) were not rewarded, instead, participants were told they would receive a portion of the reward outcome on every trial, regardless of their estimate. This was done on purpose to prevent positive RPEs due to unexpected reward when participants' predictions were correct (in our task, correct prediction implies an RPE of zero). Indeed, participants were asked to make predictions only to ensure they paid attention to outcomes in this passive-viewing, online task. It would be interesting in future work to investigate which learning conditions (e.g. Pavlovian versus instrumental) engage more collaborative versus competitive interactions between reinforcement learning and episodic memory systems.

### Positive reward prediction errors biased preference

At the end of our experiments, we investigated how RPE signals influence subsequent choice. When prompted to make choices between previously experienced scene images, participants chose both cue and outcome events linked to a higher reward outcome as well as higher (relative) value of a scene category, regardless of whether they were explicitly instructed to create these associations prior to learning (instructed memory task) or not (incidental memory task). These findings replicate previous work showing that people choose episodic events associated with higher rewards, both when these associations are formed explicitly (*Murty et al., 2016b*; *Gluth et al., 2015*) and incidentally (*Wimmer and Büchel, 2016*).

Participants thus associated both the cue and outcome scenes with the value of that scene category as well as with the specific reward outcome on that trial. Interestingly, when asked to choose between cue and outcome scenes from the same trial (where there should be no preference for either item), we found and replicated an effect (in both Experiments 1 and 2) whereby the higher the signed RPE at outcome, the more participants preferred the outcome event. This result further held when controlling for the magnitude of the outcome RPE (i.e. the unsigned outcome RPE that increased memory for outcome images), suggesting this preference was not driven by memory for large RPE events. Therefore, although signed RPEs at outcome did not modulate memory, they did predict subsequent choice, pointing to a hedonic component of the signed RPE in shaping preferences. This finding is consistent with work maintaining that RPEs drive changes in emotional or affective states (*Villano et al., 2020*; *Eldar and Niv, 2015*; *Eldar et al., 2016*; *Rutledge et al., 2014*), and we propose that this putative change in affect biased preference for the associated outcome event.

### Conclusion

Taken together, our results suggest that reward prediction errors generated both by reward-predicting cues and by reward outcomes modulate learning rate during reinforcement learning, in line with classic attentional models of learning. These signals further enhanced memory for events associated with larger unsigned prediction errors experienced at outcome (corresponding to general surprise), and larger signed prediction errors experienced at cue (corresponding to higher expected value). These findings highlight the interaction of prediction errors, potentially signaled by midbrain dopamine and locus-coeruleus norepinephrine, with mnemonic processes.

## Materials and methods

### Experimental conditions
#### Participants
We recruited participants from Amazon Mechanical Turk (MTurk): Experiment 1: 100 participants; Experiment 2, instructed memory task: 400 participants (200 for each condition); Experiment 2,

incidental memory task: 500 participants (250 for each condition). The sample size was chosen (1) based on a simulation-based power analysis revealing that at least 55 participants would give sufficient power (80% probability) to detect the effect of unsigned RPEs on memory (*Rouhani et al., 2018*), and (2) taking into account that 20% of participants typically meet one of the following exclusion criteria. Participants were excluded if they (1) had a memory score of less than 0.5 (A': Sensitivity index in signal detection; *Pollack and Norman, 1964*), or (2) missed more than three trials. More participants were recruited in Experiment 2 to test the additional effect of cue RPEs on memory. Furthermore, in the incidental version of Experiment 2, where there was no instruction to motivate remembering of the scenes, memory was worse (as could be expected), and we recruited more participants (50 per condition) to obtain similar power between the instructed and incidental memory versions of the task.

This led to a final sample of 81 participants in Experiment 1, 331 participants in Experiment 2, instructed memory task (40¢−60¢ condition: 163, 20¢−80¢ condition: 168), and 354 participants in Experiment 2, incidental memory task (40¢−60¢ condition: 168, 20¢−80¢ condition: 186). We obtained informed consent online, and participants had to correctly answer questions checking for their understanding of the instructions before proceeding; procedures were approved by Princeton University's Institutional Review Board.

## Task design

Participants each completed (1) a reward-learning task, (2) a recognition-memory task, and (3) a choice task. Before reward learning, participants completed a practice task (12 trials) to ensure they had learned the structure of the reward-learning task using different reward contingencies than what would be learned in the experimental task. In the practice trials of Experiment 1, participants experienced one reward change-point, from a mean of 30¢ to 50¢. In the practice trials of Experiment 2, in all conditions, the low-value scene category was worth 30¢ and the high-value scene category was worth 70¢, on average. Participants were additionally asked to complete a risk questionnaire (DOS-PERT; *Weber et al., 2002*) between reward learning and memory to create a 5–10 min delay between item encoding and recognition.

## Memory instructions

In the initial instructions for both Experiment 1 and the instructed memory version of Experiment 2, participants were told they would be choosing between the trial-unique images later in the experiment for a chance to win the reward associated with those events again. The aim of this choice phase was to assess learning, and informing participants about future choices was aimed at increasing attention of online participants. This instruction explicitly incentivized participants to associate images with their reward outcomes.

In the incidental memory version of Experiment 2, we tested whether our results would replicate without any incentive to remember the items. Accordingly, no instructions were given to motivate the encoding of the trial-unique images nor their association with the reward outcome on that trial. Therefore, all memory and choice results from this experiment reflect incidental encoding (see Appendix 1 for Experiment 2 instructions).

### Experiment 1 learning task

Participants learned the average value of objects in two different reward contexts, defined by background images of different cities ('Paris' and 'London'). They experienced each reward context in interleaved blocks (8 blocks total). Each block was comprised of 6 or 9 trials (60 trials total), each trial involved two trial-unique objects (120 objects in total) that were randomly assigned to each trial. On each trial, participants were first shown an object ('reward cue': 3 s), and then had up to 5 s to estimate the 'resale value of objects in that city at that time', that is, the average value of objects in that context. After submitting their answer, they saw a different trial-unique object ('reward outcome': 3 s) along with the monetary outcome associated with both objects on that trial. Participants were paid 10% of the rewards they received on every trial regardless of their estimates, in line with a Pavlovian conditioning environment.

The individual rewards associated with the object pairs fluctuated around a fixed mean (the means ranged from 10¢ to 90¢). Once or twice within each reward block, the underlying mean

changed, generating large RPEs. These 'change points' occurred once in the six-trial blocks, twice in the nine-trial blocks, and were at least three trials apart. The reward variance associated with each context provided a second source of RPEs. The variance in the high-variance context ($\sigma$-high-variance = 7¢) was twice that of the low-variance context ($\sigma$-low-variance = 3.5¢), leading participants to experience larger RPEs within the high-variance context. Participants were told that the average resale value of the 'found' objects could change within each city, but that the inherent variability in reward outcome associated with each city remained constant. Participants were encouraged to remember the rewards associated with the objects, as they were told they would be choosing between objects, and re-earning their associated rewards, later in the task.

## Experiment 2 learning task

Instead of learning the value of a single category (objects) within two reward contexts (as in Experiment 1), participants learned the value of two categories (indoor and outdoor scenes) within one reward context, thereby eliciting RPEs at cue as well as at outcome. They were told that indoor and outdoor scenes were each associated with an average value that does not change during learning, and were asked to estimate the average value of the scene category presented on every trial. As before, participants saw two different trial-unique images at reward cue and outcome, here the cue and the outcome scenes belonged to the same scene category, and images were randomly selected from each scene category.

The average value of one of the scene categories was higher than the other, and average values, as well as their variance (same for both scene categories; $\sigma$ = 15.81), remained constant throughout the experiment. In order to test a range of RPEs experienced at cue, participants learned in a reward environment where either (1) the average means of the two scene categories were close to each other ('40¢−60¢ condition': $\mu$-low-reward=40¢, $\mu$-high-reward=60¢), or (2) further apart ('20¢−80¢ condition': $\mu$-low-reward=20¢, $\mu$-high-reward=80¢). The outcomes were drawn from a predefined range centered at the above means, with the same variance between conditions ('40¢−60¢ condition': high-value scene category = 40¢−80¢, low-value scene category = 20¢−60¢; '20¢−80¢ condition': high-value scene category = 60¢−100¢, low-value scene category = 0¢−40¢), and spanned that range uniformly.

Participants completed 30 trials during learning (15 trials for each scene category; 60 trial-unique scenes). The sequence of scene-value categories (high or low scene-value categories) shown to the participant was pseudo-randomized: participants were assigned to one of eight possible sequences ensuring that no scene category was repeated consecutively more than twice, and controlling (across participants) for the number of high- and low-value scene category trials assigned to each trial number. In other words, across participants, there was a similar amount of data for both value categories on each trial.

## Learning measures

We calculated an empirical trial-by-trial outcome RPE by subtracting participants' value estimates from the reward outcome on that trial. In Experiment 2, we further calculated a cue RPE by subtracting participant's value estimates of the present reward category from the other reward category. The 'unsigned' outcome and cue RPEs were the absolute values of these measures.

We also calculated an empirical trial-by-trial learning rate directly from the Rescorla-Wagner update equation (*Rescorla and Wagner, 1972*):

$$\alpha_t = \frac{V_{t+1} - V_t}{R_t - V_t}. \tag{1}$$

We tested whether signed cue and unsigned outcome RPEs modulated this empirical learning rate.

## Recognition memory

After completing the risk questionnaire, participants were tested for their memory of the trial-unique images. They were presented with these images and asked to indicate whether they were 'old' (previously seen during learning) or 'new' (not seen during learning) as well as their confidence level for each memory judgment (from 1 'guessing' to 4 'completely certain'). In Experiment 1, the test included 72 trials: 48 old (24 from each context) and 24 new images. In

Experiment 2, the memory test included 64 trials (32 old and 32 new images). We did not test memory for every image seen during learning in order to limit fatigue and dwindling attention in online participants. However, across participants, we tested memory for the events of every learning trial by pseudo-randomizing which learning trials were probed during memory. Each participant was randomly assigned to one of four possible lists specifying which learning trials would be selected for memory testing. This ensured that events from each learning trial were probed a similar number of times in the memory test, across participants. Trial-by-trial memory scores were calculated by combining memory performance (hit versus miss) with confidence rating (from 1 = 'guessing' to 4 = 'completely certain') on old items; the score thus ranged from a 'completely certain' miss (1) to a 'completely certain' hit (8).

## Choice task

In the final phase, participants were asked to choose the more valuable image between two previously seen images (14 trials). Unbeknownst to the participants, images within each pair were either (1) both cue or outcome events from different reward pairs (in Experiment 1, these events were close in their associated reward but belonged to different variance contexts, and in Experiment 2, the events were associated with different reward outcomes; six trials), or (2) belonged to the same pair and were therefore associated with the exact same value estimate and reward (eight trials; any consistent biases in preference could not be attributable to explicit reward differences in the task).

## Reinforcement learning models

We used a simple Rescorla-Wagner model (*Rescorla and Wagner, 1972*) as our baseline model (model: 'RW'):

$$V_{t+1} = V_t + \alpha(R_t - V_t), \tag{2}$$

where a static learning rate ($\alpha$) governs the extent to which the signed RPE at outcome (computed by subtracting the current model value, $V_t$, from the reward received on that trial, $R_t$) updates the value of the next trial ($V_{t+1}$).

Following attentional models of learning (*Pearce and Mackintosh, 2010*), we investigated whether a dynamic trial-specific learning rate ($\alpha_t$) would better fit learning. We tested three distinct modulators of a trial-by-trial learning rate, separately and in combination with each other. To constrain $\alpha_t$ to be in the range of [0–1], for each model, we passed the learning rate through a sigmoid function before updating value (*Equation 2*).

First, in line with *Pearce and Hall, 1980*, we used the unsigned (absolute) outcome RPE to modulate learning rate (model: 'RW-PH'):

$$\alpha_t = \eta + \kappa(|R_t - V_c|). \tag{3}$$

Here, the unsigned outcome RPE is calculated as the difference between the reward received and the model value estimate ($V_c$). The learning rate is set as a baseline learning rate, $\eta$, plus the unsigned RPE scaled by $\kappa$. For positive values of $\kappa$, more surprising outcomes therefore lead to higher learning rates, as per the Pearce-Hall model.

Second, following *Mackintosh, 1975*, we modeled the effect of a cue RPE on learning rate (model: 'RW-M'). Note that we could only test this effect in Experiment 2 since cue RPEs exist only when there is more than one reward category. The cue RPE is the value of the present reward category (e.g. an indoor scene; $V_c$) relative to the updated value of the alternative reward category (e.g. an outdoor scene; $V_n$). The learning rate in this model is then the scaled cue RPE plus a baseline learning rate $\eta$:

$$\alpha_t = \eta + \gamma(V_c - V_n), \tag{4}$$

Therefore, for positive $\gamma$, the more one scene category is valued over the other, the higher $\alpha_t$ for trials with the more valued scene category and the lower $\alpha_t$ for trials with the less valued scene category. Since each scene category was sampled an equal number of times (without any runs exceeding two trials), we did not scale the cue RPE by the probability of either scene category occurring.

Third, given that participants should update their values less (i.e. lower their $\alpha_t$) once they've learned the average values of the reward categories, we tested a model with exponential decay of the learning rate over time (**Sutton and Barto, 1998**; model: 'RW-D'):

$$\alpha_t = \eta + N e^{-\lambda t_c}, \tag{5}$$

where $N$ is the initial value, $\lambda$ is the decay constant, and $t_c$ is the trial number for that reward category (i.e. in Experiment 2 where there were two scene categories, trial number was counted separately for each scene category).

We further tested models that included each combination of the above three learning-rate modulators. Here, we used a single baseline ($\eta$) and added each effect in the learning rate for all of the following models: A model that combines the unsigned outcome RPE and signed cue RPE effects on learning rate (model: 'RW-PH-M'):

$$\alpha_t = \eta + \kappa(|R_t - V_c|) + \gamma(V_c - V_n), \tag{6}$$

A model that combines the unsigned outcome RPE and decay effects on learning rate (model: 'RW-PH-D'):

$$\alpha_t = \eta + \kappa(|R_t - V_c|) + N e^{-\lambda t_c}, \tag{7}$$

A model that combines the signed cue RPE and decay effects on learning rate (model: 'RW-M-D'):

$$\alpha_t = \eta + \gamma(V_c - V_n) + N e^{-\lambda t_c}, \tag{8}$$

And finally, a model that combines all three effects (model: 'RW-PH-M-D'):

$$\alpha_t = \eta + \kappa(|R_t - V_c|) + \gamma(V_c - V_n) + N e^{-\lambda t_c}. \tag{9}$$

## Model fitting and comparison

All models were fit to each participant's value estimates by finding parameters that maximize the log likelihood of the participant value estimates. The likelihood was calculated assuming a Gaussian distribution around the model value, with variance equal to the average empirical difference between model values and participant estimates ($\sigma^2$). This is equivalent to linear regression of the value estimates on the model values, giving a log likelihood:

$$LL = -n_{data}\left[ ln\left( \sqrt{2\pi\sigma^2} \right) + 0.5 \right], \tag{10}$$

where $n$ is the number of trials fit. To maximize log likelihood we used MATLAB's *fmincon* function. We constrained parameter values within the following ranges: $\alpha \in [0,1]$, $\eta \in [-10,10]$, $\kappa \in [-20,20]$, $\gamma \in [-20,20]$, $N \in [-15,15]$, $\lambda \in [0,10]$. Note, however, that the trial-by-trial learning rate was always passed through a sigmoid function ($x_t$ = input), and was therefore between 0 and 1:

$$\alpha_t = \frac{1}{1 + e^{-x_t}}. \tag{11}$$

Values were initialized to 50¢, and in Experiment 1, were re-initialized at the beginning of each reward context. Each fit was run 30 times with different random initial parameter values.

Since all our models were nested (with additional parameters further modulating the RW-learning rate), we compared models using the likelihood-ratio test, across subjects (**Pickles, 1985**). To verify that our data can arbitrate between these models, we performed model recovery on simulated data generated by randomly sampling 100 parameter settings from Experiment 2 (including sampling the Gaussian noise translating model value to predicted value). From these simulated data we calculated empirical trial-by-trial learning rates (as in **Equation 1**). We then tested whether the model generating said learning rates was the best fit for them, by fitting all models to each dataset. We concentrated specifically on modeling learning rates, since the only differences between the models were in how they determined trial-by-trial learning rates. We then compared model recovery using the conservative Bayesian information criterion (BIC; **Schwarz, 1978**), to calculate a confusion matrix demonstrating the proportion of

simulations fit best by the true model (*Wilson and Collins, 2019*). The models were sufficiently recovered, validating model comparison (*Figure 2—figure supplement 2B*). Code for model fitting and recovery in 'models_RL_matlabCode' at https://github.com/ninarouhani/2021_RouhaniNiv (*Rouhani, 2021*; copy archived at swh:1:rev:fa15d035dc4033ebad03f48dbd5c75b0c4d76c40).

## Mixed-effects modeling

We used mixed-effects modeling to test hypotheses throughout the paper (lme4 package in R; *Bates et al., 2015*). We treated participant as a random effect for both the slope and the intercept of each fixed effect; however, if the model did not converge, we incrementally simplified the random effect structure (i.e. by taking out interactions, then the slope of each effect), until convergence was achieved (the simplest structure only modeled participant intercept as a random effect; for model specifications, see 'analysis&figures.ipynb' at https://github.com/ninarouhani/2021_RouhaniNiv/).

## Hierarchical model of memory

We ran a hierarchical regression model to better characterize the effects of unsigned and signed RPES, as well as their relative influence, on memory for cue and outcome events. This model performed full Bayesian inference over the effects of interest with Hamiltonian Monte Carlo sampling, simultaneously estimating subject and group-level posterior distributions (Stan; *Carpenter et al., 2017*). We included all putative RPE signals of interest in predicting memory score: signed RPE signal at outcome, unsigned RPE signal at outcome, as well as an intercept and a nuisance variable that captured overall differences in memory for cue versus outcome events. We also included signed and unsigned RPE signals at cue for Experiment 2. Subject-level parameter distributions were drawn from group-level, standard normal distributions, and scaled by a gamma distribution (1,0.5). The response variable (memory score) was modeled with a normal distribution and fit with a single Gaussian noise parameter across all participants. All RPE regressors were centered and standardized. We report the median (M) of the posterior parameter distributions as a measure of centrality, and the highest density interval (HDI) as a measure of uncertainty around the parameter estimate; by default, HDI returns the 89% credible interval (which is recommended as a more stable interval for sample sizes less than 10,000; *Kruschke, 2014*; *Makowski et al., 2019*). Code for Stan models in 'models_memory_stanCode' at https://github.com/ninarouhani/2021_RouhaniNiv/.

## Acknowledgements

We thank Angela Radulescu and Isabel Berwian for helpful comments. This work was supported by grant W911NF-14-1-0101 from the Army Research Office (YN), grant R01MH098861 from the National Institute for Mental Health (YN), grant R21MH120798 from the National Institute of Health (YN) and the National Science Foundation's Graduate Research Fellowship Program (NR).

## Additional information

### Funding

| Funder | Grant reference number | Author |
| --- | --- | --- |
| Army Research Office | W911NF-14-1-0101 | Yael Niv |
| National Institute of Mental Health | R01MH098861 | Yael Niv |
| National Science Foundation | Graduate Student Fellowship | Nina Rouhani |
| National Institutes of Health | R21MH120798 | Yael Niv |

The funders had no role in study design, data collection and interpretation, or the decision to submit the work for publication.

## Author contributions
Nina Rouhani, Conceptualization, Data curation, Software, Formal analysis, Funding acquisition, Validation, Investigation, Visualization, Methodology, Writing - original draft, Project administration, Writing - review and editing; Yael Niv, Conceptualization, Resources, Supervision, Funding acquisition, Validation, Methodology, Writing - original draft, Writing - review and editing

## Author ORCIDs
Nina Rouhani https://orcid.org/0000-0003-2814-0462
Yael Niv http://orcid.org/0000-0002-0259-8371

## Ethics
Human subjects: We obtained informed consent online; procedures were approved by Princeton University's Institutional Review Board (IRB #4452).

## Decision letter and Author response
Decision letter https://doi.org/10.7554/eLife.61077.sa1
Author response https://doi.org/10.7554/eLife.61077.sa2

# Additional files
## Supplementary files
• Transparent reporting form

## Data availability
All data files and code for models, analysis and figures are publicly available at https://github.com/ninarouhani/2021_RouhaniNiv copy archived at https://archive.softwareheritage.org/swh:1:rev:fa15d035dc4033ebad03f48dbd5c75b0c4d76c40/.

The following dataset was generated:

| Author(s) | Year | Dataset title | Dataset URL | Database and Identifier |
|---|---|---|---|---|
| Rouhani N, Niv Y | 2021 | 2021_RouhaniNiv | https://github.com/ninarouhani/2021_RouhaniNiv | github, 2021_RouhaniNiv |

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

## Appendix 1

### Experiment 2 instructions (initial instructions prior to reward learning/encoding)
Differences between instructions for the instructed and incidental memory versions of the task noted below

Welcome!

In this experiment, you will be viewing a bunch of photographs (organized in pairs of photos), some of outdoor scenes and some of indoor scenes.

Each pair of photos has a value between 0 and 1 dollar – you will be presented with the photos, and see (and win) the money value of each.

One type of photograph - either photos of indoor scenes or photos of outdoor scenes - is worth more money overall making it the winner.

To learn about the values of indoor and outdoor images, you will get a chance to watch the computer choose pairs of indoor or outdoor photographs.

For each pair that the computer chooses, you will see one of the photos, and be asked to estimate from 0 to 100 cents, how much you think on average pairs of scenes of its type (indoor or outdoor) are worth. You will have 5 s to enter your estimate.

You will not know the answer at first, but please make your best guess of what the average value of a pair of scenes of this type is. To submit your answer, press enter.

After submitting your estimate, the true value of the pair will appear along with the second photo belonging to the chosen pair. Although different pairs of photos have different values, their worth is solely determined by the general value of that type of scene (indoor or outdoor). The average value of each type of scene does not change over this whole task.

Pay attention to the money reward you get, so you can update your estimate of the average value of outdoor and indoor scenes

Instructed memory: IMPORTANT: You will be paid a portion of the worth of the photos you see. You will receive approximately 1 cent for every 10 cents rewarded. Additionally, you will be able to use the reward value of each pair to choose between the same photos later in the task, and win their values. The more you win, the more you will be paid at the end of this HIT.

Incidental memory: IMPORTANT: These outcomes are real. You will be paid a portion of the outcomes of the photos you see. You will receive approximately 1 cent for every 10 cents rewarded.

The order of the task is as follows: (1) you will first see one of the two photos that the computer chose, and determine whether it's an indoors or outdoors scene, (2) you will give an estimate of how much this type of photo (indoor or outdoor) is worth overall, and after submitting your answer, (3) you will see how much the pair is actually worth along with the second photo belonging to the pair.

Please pay close attention! You will have 3 s to view the first photo in a pair, 5 s to submit your estimate of its average value and 3 s to see the second photo and the value of the chosen pair.

After looking through all of the photos, you will then be asked to indicate the 'winner'. In other words, you will indicate which type of scene (indoor or outdoor) has the higher value on average.

To receive full payment, you will need to complete this experiment and submit it. In the case of an event that precludes you from completing the experiment, please return the HIT and do not submit it. In that case, please e-mail nrouhani@princeton.edu to be compensated for the time you did spend on the task.

Comprehension questions (must be answered correctly before starting the task)

Before starting the experiment, please answer the following questions and click submit. Once you've answered all the questions correctly, the task will automatically load. if you do not answer correctly, you will see the instructions again.

Which statement is true about indoor versus outdoor photos?

a. One type of scene (either indoor or outdoor) is more valuable, on average
b. Scenes can take on any value, and whether it is indoor or outdoor does not matter
c. Photos within a pair can belong to different scene types

Which statement is true about the value of each pair?

a. The value of each pair dosen't affect your winnings

b. The value of each pair represents the amount you are winning
c. The value of each pair can change without warning

What determines the value of each pair?

a. The average value of outdoor and indoor scenes (although pair may take on different values from each other)
b. The quality of the photos in the pair
c. The attractiveness of the scenes in the pair

## Practice instructions

You will now complete a short practice run before starting the real task.

Remember:

1. On each turn, the computer will choose a pair of photos and you will estimate how much, on average, you think the type of scene on the screen (indoor or outdoor) is worth.

2. After submitting your estimate (by clicking or pressing enter), you will see the second photo and the true value of that pair, and will win a portion of the reward.

Instructed memory: 3. Note each pair and its value - you will be choosing the same photos later in the task.

Incidental memory: (this point is deleted)

4. At the end of this task, you will indicate whether indoor or outdoor scenes were the 'winner'.

Please do not take notes, it is not necessary for successful performance, and will ruin the experiment.

