## [Decision Letter]

**Acceptance summary:**

This manuscript is of broad interest for psychologists and neuroscientists. The finding that error signals during reinforcement learning affect not only the rate of learning but also the degree to which events occurring during learning are committed to memory, and that these effects are independent of whether learning is instructed or incidental, has important implications for our understanding of how different learning systems interact. These conclusions are supported by strong behavioral and computational modeling data.

**Decision letter after peer review:**

Thank you for submitting your article "Signed and unsigned reward prediction errors dynamically enhance learning and memory" for consideration by *eLife*. Your article has been reviewed by three peer reviewers, one of whom is a member of our Board of Reviewing Editors, and the evaluation has been overseen by Christian Büchel as the Senior Editor. The reviewers have opted to remain anonymous.

The reviewers have discussed the reviews with one another and the Reviewing Editor has drafted this decision to help you prepare a revised submission.

Summary:

The computational rules through which the brain prioritizes experiences for storage in episodic memory are an active area of research, and one where there has been considerable disagreement in the literature. Here Rouhani and Niv rectify seemingly contradictory results from previous studies by showing that effects of rewarding and surprising outcomes on memory occur at different times in the course of a decision task, and they go on to link them to longstanding learning signals that have played instrumental roles in behavioral psychology.

Reviewers agreed that the findings are timely and interesting. There were extensive discussions among reviewers about critical aspects of the current task design that may make it difficult to relate the current results to previous findings. Specifically, the instructions given to participants encourage explicitly evaluating the stimuli and remembering them for later use. The concern is that even though the authors claim to resolve conflicting findings about modulators of memory, this is difficult to achieve with the current design because it is too different from existing work on reward-modulated memory. To better integrate the current findings with previous work, it would be important to replicate experiment 2 using instructions that do not encourage explicit memorization and allow for more incidental memory encoding.

Essential revisions:

1) There is a qualitative distinction between the current intentional encoding-for-later-use task and the overwhelmingly more common incidental memory tasks that have been used for decades, and to which the authors are trying to connect to. In general, even though the authors claim to resolve some apparently contradictory predictions about modulators of memory, they cannot do that with this design because it is too different from existing work on reward-modulated memory. Their instructed design makes it more akin to the reward-motivated memory work started by Adcock et al., 2006, where participants were incentivized to remember pictures based on cued value. The closest parallel to the current report is the work by Murty et al., 2016; see their Table 1, Experiment 3). There, participants were instructed to remember stimuli for later in the experiment (although remembering associated outcomes was not explicitly instructed). While not the focus of their study, they report no effect of reward on later memory. The current results could be part of the start of a different research direction (building on null findings from Murty et al.) on the relationship between strategic encoding-for-later-use, reward, and memory. It is unclear whether that direction would shed light on how memory is actually used outside of the lab, and such a re-framing would require a very significant re-framing of the current manuscript. Focusing on what the instructed encoding of pictures and outcomes does to memory, what do the results suggest? In these situations, it may be a rational strategy to pay greater attention to and prioritize the encoding of highly variant outcomes (surprising positive and negative deviations from expectations) if participants are trying to remember these associations for later choices. Thus, the reported modulation of memory by unsigned prediction error may be an expected consequence of instructing participants to remember associations between pictures and outcomes. To overcome these issues, reviewers would like to see an exact replication of experiment 2 without explicit instructions to memorize the items. Given data are collected online, this should be possible despite potential lab closings due to COVID-19.

2) Reviewers were not completely convinced by the model-fitting, although they felt it plays a relatively minor role for the current study, as it is unclear whether the Mackintosh signal needs to affect decision making behavior in order to affect longer term memory storage. Regardless, the model selection would be more convincing if it included the following additional information: 1) a confusion matrix demonstrating that at least one of the model comparison tools used can reliably recover the correct model, 2) posterior predictive checks showing that the winning model can capture basic learning curves, and 3) parameter fits from the winning model that make clear systematic contributions of the Mackintosh and Pearce Hall terms to positive (or negative) adjustments to the learning rate. If there is no systematic sign for Mackintosh parameter, it would be useful to know whether the sign of fits per subject corresponds to memory difference for images on positive versus negative RPE trials.

3) An additional concern is whether the proposed cue RPE modulator is actually specific to the cue RPE. On each trial, there are two factors that are highly correlated with cue RPE: the estimated value and the feedback value. In the current data, the benefit of including the Mackintosh effect on the model fits suggests that the effect comes on after values begin to be differentiated, supporting a cue RPE influence. However, an alternative model where the effect exists already at the start of learning was not evaluated. The task is very short – there are only 30 trials (15 per category) in Experiment 2. It is reasonable to assume that some initial period is necessary for participants to adjust to the task requirements / establish a “task-set” guiding their behavior, which here could cover a meaningful proportion of trials. Even if fits are better for a delayed onset of the Mackintosh effect, this could just be due to initial task adjustment adding noise to initial trials. If the participants engaged in an additional round of learning (over a new set of two types of stimuli), or if there was an extended practice session before the onset of the currently unfamiliar task, the authors could determine whether their observed result is actually due to acquisition of a cue value (leading to cue RPEs). If the observed effect is instead related to the relative estimated value or relative feedback magnitude, these would have a rapid influence on learning (once a few events are observed to roughly determine the feedback range, similar to scaling effects on dopamine RPEs). Following from this, if the feedback value is driving the observed effect, this might be due to an influence of memory for specific preceding feedback value. Basing next-trial estimates on memory for preceding trial feedback would lead to a higher estimated learning rate. This memory effect could be due to short-term working memory, given the rapid trial repetition (Collins et al.), or episodic memory.

---

## [Author Response]

Essential revisions:1) There is a qualitative distinction between the current intentional encoding-for-later-use task and the overwhelmingly more common incidental memory tasks that have been used for decades, and to which the authors are trying to connect to. In general, even though the authors claim to resolve some apparently contradictory predictions about modulators of memory, they cannot do that with this design because it is too different from existing work on reward-modulated memory. Their instructed design makes it more akin to the reward-motivated memory work started by Adcock et al., 2006, where participants were incentivized to remember pictures based on cued value. The closest parallel to the current report is the work by Murty et al., 2016; see their Table 1, Experiment 3). There, participants were instructed to remember stimuli for later in the experiment (although remembering associated outcomes was not explicitly instructed). While not the focus of their study, they report no effect of reward on later memory. The current results could be part of the start of a different research direction (building on null findings from Murty et al.) on the relationship between strategic encoding-for-later-use, reward, and memory. It is unclear whether that direction would shed light on how memory is actually used outside of the lab, and such a re-framing would require a very significant re-framing of the current manuscript. Focusing on what the instructed encoding of pictures and outcomes does to memory, what do the results suggest? In these situations, it may be a rational strategy to pay greater attention to and prioritize the encoding of highly variant outcomes (surprising positive and negative deviations from expectations) if participants are trying to remember these associations for later choices. Thus, the reported modulation of memory by unsigned prediction error may be an expected consequence of instructing participants to remember associations between pictures and outcomes. To overcome these issues, reviewers would like to see an exact replication of experiment 2 without explicit instructions to memorize the items. Given data are collected online, this should be possible despite potential lab closings due to COVID-19.

We thank reviewers for this suggestion and have now replicated our main results from Experiment 2 in a task where we did not motivate encoding of the scenes (N = 354). We have further reframed our paper to include both of these Experiment 2 datasets, one encouraging explicit memory (original data) and the other incidental memory (we include instructions for each version of Experiment 2 in Appendix 1). Our approach was to model learning behavior across the instructed and incidental memory versions of the task (i.e., combining datasets) since learning instructions were the same, and separately analyze the memory and choice results between these two versions. We did, however, find unanticipated differences in learning performance between the two versions of Experiment 2, which we report in the manuscript and further link to memory results.

To begin, we replicated our main findings supporting the putative neurobiological mechanisms we describe in this manuscript: unsigned RPEs at outcome increase memory for trial-unique images presented at reward outcome (*β* = 0.11, *t* = 3.01, *p* = 0.003; Figure 4D) and signed RPEs at cue increase memory for images presented at cue (*β* = 0.09, *t* = 3.04, *p* = 0.002; Figure 4E).

In other words, in a task where no instructions were given to remember the items, nor to associate them with their value, incidental memory was enhanced for both images associated with more surprising rewarding outcomes and more rewarding cue events. We also found, in the incidental version, that *unsigned* RPEs at cue increased memory for cue images (*β* = 0.10, *t* = 2.93, *p* = 0.003; Figure 4C), an effect that had only been trending in the explicit version. This effect shows that the more participants had separated the values of the two reward cues, the better they were remembering the cue image, additionally supporting the increasing memory for cue events over learning. We did not replicate signed cue RPE’s enhancement of outcome items in the incidental version (Figure 4E, purple distribution), possibly because that effect was due to a more explicit encoding strategy where participants were encoding both the cue and outcome items associated with the higher reward category.

As in the instructed version, we also found an overall increase in memory for cue images over learning in the incidental version. However, when analyzing the two learning conditions of Experiment 2 separately (recall that this experiment included a harder condition where the means of two reward categories were 40¢ and 60¢, and an easier condition where the means were further apart, 20¢ and 80¢), we only found this overall increase in cue memory in the easier condition (Figure 3D, E).

We note here that although we did not find that the learning behavior from the new experiment influenced our modeling results (i.e., the same model was still the best fitting), we did find learning performance to be generally worse in the incidental memory version of the task (even though the learning instructions were exactly the same). We now report these results under “Learning behavior in the experimental conditions of Experiment 2” (and note that the incidental version was run during the COVID pandemic). We therefore examined whether these learning differences could account for overall memory differences between the incidental and instructed memory versions using an individual difference approach. Across all conditions of Experiment 2, we tested whether individual learning performance predicted the increase in cue memory as a function of trial number during learning. Indeed, we found that better learners also showed a greater increase of memory for cue images, but not outcome images, over learning (Figure 3—figure supplement 1). This relationship was particularly strong in the harder learning condition of the incidental memory task – that is, in the replication experiment, where we did not observe an overall increase in cue memory, suggesting that the latter null result may be driven by the overall prevalence of worse learners (while the few better learners still show the effect). This analysis provided an additional between-subjects link connecting a gradual increase in memory for cue images to more successful learning.

We also replicated all of the previous choice findings. Even though participants were not instructed to encode the association between images and their outcomes, they still preferred cue and outcome images associated with higher reward outcomes and values at the end-of-experiment surprise choice test. This indicates that they were associating both cue and outcome images with the reward outcome and value experienced on that trial (Figure 5C, D). Moreover, as before, when choosing between a cue and an outcome image from the same pair (where there should be no preference for either image), participants “irrationally” preferred the outcome image the more positive the signed RPE at outcome (Figure 6C).

Given the change from a short report to a full paper, we now include this new version of the experiment, as well as a fuller analysis of both learning and memory results across all experiments throughout the manuscript. We also provide a Discussion section where we relate these results to the larger literature investigating the effects of reward learning signals on subsequent memory (including all of the papers suggested in this review).

2) Reviewers were not completely convinced by the model-fitting, although they felt it plays a relatively minor role for the current study, as it is unclear whether the Mackintosh signal needs to affect decision making behavior in order to affect longer term memory storage. Regardless, the model selection would be more convincing if it included the following additional information: 1) a confusion matrix demonstrating that at least one of the model comparison tools used can reliably recover the correct model, 2) posterior predictive checks showing that the winning model can capture basic learning curves, and 3) parameter fits from the winning model that make clear systematic contributions of the Mackintosh and Pearce Hall terms to positive (or negative) adjustments to the learning rate. If there is no systematic sign for Mackintosh parameter, it would be useful to know whether the sign of fits per subject corresponds to memory difference for images on positive versus negative RPE trials.

Thank you for these important suggestions. With respect to 1, we validated model comparison by calculating a confusion matrix, *p* (fit model|true model), using BIC for all models used in Experiment 2 (which also comprises all models used in Experiment 1, Figure 2—figure supplement 2B). Given that all tested models were nested within the learning rate, we simulated trial-by-trial learning rates from the parameters fit to values, and directly fit these learning rates. As indicated by the high proportion of recovered simulations, this procedure validated our model comparison. We include code for the model-validation analysis in “models_RL_matlabCode” in the github repository. Since our experiments afforded measurement of trial-by-trial learning rate, we were also able to compare the models by fitting the empirical learning rates. This analysis resulted in the same pattern of results and winning model as when fitting values (we do not report these results in the paper since for this analysis we could not include trials where the prediction error was 0 leading to a learning rate of infinity, which substantially reduced the number of valid trials for many subjects).

With respect to 2, we had represented the model-simulated values (which function as posterior predictive checks) in the original learning plots, in Figure 2A-C of the manuscript. Here, the black dots on the learning curves represent the value estimates generated by the winning model for each experiment.

Regarding 3, the winning model includes several parameters that interact with each other to predict trial-by-trial values. We plot these individually (Author response image 1), but believe that showing the parameter distributions in the manuscript would not help readers understand how the model predicts behavior. Instead, we chose to include the trial-by-trial distributions for learning rates generated by the winning model of Experiment 2 (Figure 2—figure supplement 2A). The takeaway here is that dynamically changing learning rates are larger and more variable at the beginning of learning, but decrease in size over the course of learning. Because of the interactions between model parameters, we did not correlate individual parameter estimates with memory for cue or outcome images as such correlations may be difficult to interpret.

**Author response image 1. sa2fig1:** Model validation simulations. 'RW': Rescorla-Wagner, 'PH': Pearce-Hall, 'M': Mackintosh, 'D': Decay. Distribution of parameter estimates from the winning model in Experiment 1 ('RW-PH-D') and Experiment 2 ('RW-PH-M-D').

3) An additional concern is whether the proposed cue RPE modulator is actually specific to the cue RPE. On each trial, there are two factors that are highly correlated with cue RPE: the estimated value and the feedback value. In the current data, the benefit of including the Mackintosh effect on the model fits suggests that the effect comes on after values begin to be differentiated, supporting a cue RPE influence. However, an alternative model where the effect exists already at the start of learning was not evaluated. The task is very short – there are only 30 trials (15 per category) in Experiment 2. It is reasonable to assume that some initial period is necessary for participants to adjust to the task requirements / establish a “task-set” guiding their behavior, which here could cover a meaningful proportion of trials. Even if fits are better for a delayed onset of the Mackintosh effect, this could just be due to initial task adjustment adding noise to initial trials. If the participants engaged in an additional round of learning (over a new set of two types of stimuli), or if there was an extended practice session before the onset of the currently unfamiliar task, the authors could determine whether their observed result is actually due to acquisition of a cue value (leading to cue RPEs). If the observed effect is instead related to the relative estimated value or relative feedback magnitude, these would have a rapid influence on learning (once a few events are observed to roughly determine the feedback range, similar to scaling effects on dopamine RPEs). Following from this, if the feedback value is driving the observed effect, this might be due to an influence of memory for specific preceding feedback value. Basing next-trial estimates on memory for preceding trial feedback would lead to a higher estimated learning rate. This memory effect could be due to short-term working memory, given the rapid trial repetition (Collins et al.), or episodic memory.

Thank you for these critical points. First and foremost, we made a mistake in not including information about the practice block that always preceded learning. In order to familiarize participants with the task-set, as you mention, we had participants complete a practice section identical to the learning block where they learned the means of reward categories, although with different underlying means than the learning task and fewer trials. The same practice trials were given in each experiment to avoid differentially biasing participants between conditions. We now include these details in the Materials and methods section.

Before reward learning, participants completed a practice task (12 trials) to ensure they had learned the structure of the reward-learning task using different reward contingencies than what would be learned in the experimental task. In the practice trials of Experiment 1, participants experienced one reward change-point, from a mean of 30¢ to 50¢. In the practice trials of Experiment 2, in all conditions, the low-value scene category was worth 30¢ and the high-value scene category was worth 70¢, on average.

As to your point about participants potentially using their working memory of previous outcomes for subsequent value estimates, we now include empirical learning rate plots for each experiment as a supplementary figure that sheds light on this issue (Figure 2—figure supplement 1). If participants were using the last reward outcome as their subsequent estimate for that reward category, learning rates would be close to 1. We do not see such high learning rates on average (except for learning rates after change points in Experiment 1, where we would expect a very high learning rate based on prior literature). We note that empirical learning rates in Experiment 2, which includes the cue RPE signal (Figure 2—figure supplement 1B, C), were low (~0.5 or less, on average) from the very first trials of the experiment.